# LEARNING GAIN MAP FOR INVERSE TONE MAPPING

**Yinuo Liao, Yuanshen Guan, Ruikang Xu, Jiacheng Li, Shida Sun, Zhiwei Xiong***
University of Science and Technology of China, Hefei, China
`{yinuoliao,ysguan,xurk,jclee,sdsun}@mail.ustc.edu.cn,`
`zwxiong@ustc.edu.cn`

## ABSTRACT

For a more compatible and consistent high dynamic range (HDR) viewing experience, a new image format with a double-layer structure has been developed recently, which incorporates an auxiliary Gain Map (GM) within a standard dynamic range (SDR) image for adaptive HDR display. This new format motivates us to introduce a new task termed Gain Map-based Inverse Tone Mapping (GM-ITM), which focuses on learning the corresponding GM of an SDR image instead of directly estimating its HDR counterpart, thereby enabling a more effective up-conversion by leveraging the advantages of GM. The main challenge in this task, however, is to accurately estimate regional intensity variation with the fluctuating peak value. To this end, we propose a dual-branch network named GMNet, consisting of a Local Contrast Restoration (LCR) branch and a Global Luminance Estimation (GLE) branch to capture pixel-wise and image-wise information for GM estimation. Moreover, to facilitate the future research of the GM-ITM task, we build both synthetic and real-world datasets for comprehensive evaluations: synthetic SDR-GM pairs are generated from existing HDR resources, and real-world SDR-GM pairs are captured by mobile devices. Extensive experiments on these datasets demonstrate the superiority of our proposed GMNet over existing HDR-related methods both quantitatively and qualitatively. The codes and datasets are available at https://github.com/qtlark/GMNet.

## 1 INTRODUCTION

High dynamic range (HDR) images are widely employed in the media and film industries, offering a realistic visual experience with vivid details. To enable a more compatible and consistent HDR image display, a new image format with a double-layer structure has been developed recently (Apple, 2021; Adobe, 2024; Google, 2024; ISO, 2024). This format stores a standard dynamic range (SDR) image and an auxiliary Gain Map (GM), enabling HDR image adaptation across various devices. Specifically, it allows the SDR image to be directly displayed on legacy devices, while applying the GM to the SDR image for HDR display on modern devices as shown in Fig. 1 (a).

Motivated by this novel double-layer format, we introduce a new Inverse Tone Mapping (ITM) task to up-convert existing SDR images to their HDR version, which is termed Gain Map-based Inverse Tone Mapping (GM-ITM). It targets GM estimation instead of direct HDR image prediction, fully exploiting the following advantages of the GM: (1) Simplified construction: The GM stores pixel-level dynamic range information in a compact 8-bit depth image, with a resolution typically reduced to 1/4 or 1/16 of the original SDR image. (2) Balanced distribution: The GM exhibits a more balanced pixel-value distribution compared with the HDR image, as shown in Fig. 1 (b). (3) Detail preservation: As shown in the top row of Fig. 1 (c), logarithm-encoded GM alleviates the over-compression problem, preserving more contrast and texture details in highlight regions.

In addition to leveraging the inherent advantages of the GM, the characteristics of the GM-ITM further enhance its superiority in up-conversion compared to previous learning-based HDR-related tasks (Yao et al., 2023; Huang et al., 2023; Li et al., 2022; Liu et al., 2023). First, GM-ITM shifts the target from the HDR image to the intermediary GM, simplifying the learning process since predicting the transformation from input to output is simpler than directly predicting the output (Gharbi

---

*Corresponding author.

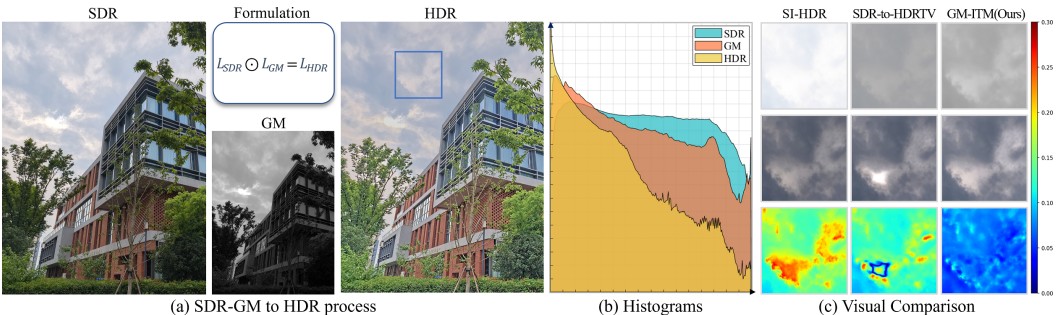

Figure 1: (a) Schematic of the SDR-GM to HDR process and simplified formulation of Eq. 3. The GM records pixel-wise dynamic range information of the corresponding SDR image and can be used to restore HDR through element-wise multiplication. (b) Histograms of SDR, GM, and HDR, with logarithmic statistics computed across 100 images. The GM illustrates a more balanced distribution compared to the long-tailed distribution of the HDR image, making it a more tractable learning target. (c) Visual comparison of methods (Chen et al., 2021a; Xu et al., 2022) for different HDR-related tasks. The top row demonstrates the supervision targets, while the rows below show the estimation results and the error maps, in which our method achieves the best visual experience.

et al., 2017). Second, GM-ITM is implemented during the display process, enabling a more flexible up-conversion compared to previous tasks that aim at irradiance estimation of the scenes (Khan et al., 2019; Kim et al., 2021) or up-conversion of the images (Cheng et al., 2022; Liu et al., 2024).

While the GM brings benefits to the ITM process, it also presents challenges due to its unique characteristics, particularly in estimating regional intensity variation with the fluctuating peak value. To address this problem, we propose GMNet, a network with two branches fitting the locality and globality of the GM respectively, in which the Local Contrast Restoration (LCR) branch aims at recovering pixel-wise spatial information, and the Global Luminance Estimation (GLE) branch focuses on predicting image-wise luminance information. Additionally, considering the intrinsic relationship between the locality and globality of the GM, we introduce a squeeze module for global guidance extraction, and propose spatial-aware and channel-wise modulation modules to align the LCR branch with the GLE branch, ensuring a precise and consistent up-conversion across both aspects. Based on the above designs, GMNet not only overcomes the difficulties in learning GM, but also effectively utilizes the intrinsic connections behind these difficulties, thus achieving superior performance.

Furthermore, we build synthetic and real-world datasets to facilitate future research on the GM-ITM task. Specifically, the synthetic dataset consists of SDR-GM pairs derived from HDRTV-standard videos, while the real-world dataset comprises high-resolution SDR-GM pairs captured by mobile devices. The datasets demonstrate extensive diversity, covering both daytime and nighttime scenes, as well as indoor and outdoor settings. The quantitative and qualitative experiments on these datasets indicate that our method significantly outperforms the existing HDR-related methods.

Contributions of this paper can be summarized as follows:

- Motivated by a new double-layer HDR image format, we introduce a new task named GM-ITM to achieve a more effective up-conversion.
- We propose a dual-branch network tailored for GM-ITM, exhibiting superior performance in both qualitative and quantitative evaluations to existing solutions.
- To facilitate further research along this line, we build both synthetic and real-world datasets consisting of high-resolution and diverse SDR-GM pairs.

## 2 RELATED WORK

Learning-based HDR-related tasks have been a long-standing research topic. However, many of them are similar in name but differ in connotation, as shown in Tab. 1. In this section, we will distinguish the scopes of these tasks and introduce related studies.

**Single-Image HDR reconstruction (SI-HDR).** Due to the limited dynamic range of camera sensors, image quality degrades in extremely bright or dark scenes, motivating SI-HDR methods to

| Task | Input | | | Output | | |
|------|-------|------|-------|--------|------|-------|
| | Type | OETF | Gamut | Type | OETF | Gamut |
| SI-HDR | LDR | Linear | Specific | HDR | Linear | Specific |
| HDR-style enhancement | LDR | Gamma | BT.709 | SDR | Gamma | BT.709 |
| SDR-to-HDRTV | SDR | Gamma | BT.709 | HDR | PQ/HLG | BT.2020 |
| GM-ITM | SDR | Gamma | P3 | GM | Logarithm* | N/A |
| | | | | HDR | Linear | P3 |

Table 1: Comparison of the learning-based HDR-related tasks. The table lists typical values, yet there may be alternatives in practice. The *LDR* and *SDR* essentially represent low dynamic range content, but the LDR image is derived from the camera ISP, while the SDR image is degraded from the HDR content (Chen et al., 2021b). The color gamut of SI-HDR is device-specific, while other tasks follow the definition of BT.709 (ITU-R, 2015b), BT.2020 (ITU-R, 2015a) and P3 (DCI, 2005; SMPTE, 2010), and the gamut is not applicable (N/A) for the GM. *The opto-electronic transfer function (OETF) of GM being undefined, we adopt the logarithm encoding function as a substitute.

restore HDR irradiance from the LDR source. The SI-HDR methods can be broadly divided into two branches (Wang & Yoon, 2021), one of which is the direct mapping approach. Eilertsen et al. (2017) utilize an end-to-end CNN to recover details in over-exposed regions. HDRUNet (Chen et al., 2021a) uses a spatially dynamic encoder-decoder network to learn the LDR-to-HDR mapping. KUNet (Wang et al., 2022) introduces a knowledge-inspired block to capture global information for HDR reconstruction. DCDR-UNet (Kim et al., 2024) introduces the deformable block to better restore lost details in the over-exposed region regardless of the size. The other branch is stack-based approach. Endo et al. (2017) propose the first deep-learning-based approach that reconstructs the HDR image by merging estimated bracketed LDR images. Lee et al. (2018a) generate HDR images from multi-exposure stack using a conditional generative adversarial network. Zhang et al. (2023) estimate two exposures to reconstruct HDR radiance from a single image. In general, SI-HDR methods focus on reconstructing missing details and recovering scene irradiance.

**HDR-style enhancement.** For a better visual experience within a limited dynamic range, HDR-style enhancement methods upgrade LDR images to SDR images in HDR-like view to improve perceptual quality. In previous studies, HDRNet (Gharbi et al., 2017) predicts locally affine model coefficients in bilateral space, achieving pleasing results on the HDR plus dataset (Hasinoff et al., 2016). Zheng et al. (2021) propose a dual-path network that reconstructs high-quality content and chromatic features using guided bilateral up-sampling. These enhancement methods aim to improve the visual quality of LDR images, but without dynamic range broadened in fact.

**SDR-to-HDRTV up-conversion.** The popularity of HDR monitors calls for SDR-to-HDRTV up-conversion methods to broaden the dynamic range and the color gamut, upgrading the existing SDR source to the HDRTV version. Kim & Kim (2019) first investigate this by jointly learning super-resolution and inverse tone-mapping. FMNet (Xu et al., 2022) introduces a frequency-aware modulation block to reduce structural distortions. He et al. (2022) propose a two-stage method using hierarchical feature modulation and dynamic context feature transformation. ITM-LUT (Guo et al., 2023b) combines LUT with AI, achieving an efficient display-end ITM. However, these methods struggle in extremely bright regions due to over-compressed learning targets, as shown in Fig. 1 (c).

**Gain Map-based Inverse Tone Mapping.** The SDR image is rendered to SDR display for legacy devices, while upgrades to HDR display with the help of GM (Canham et al., 2024). It inspires GM-ITM methods to predict GM for adapting SDR to HDR display. Compared to SI-HDR, which aims to reconstruct scene irradiance, GM-ITM is more akin to SDR-to-HDRTV up-conversion, as they are both ITM tasks. However, GM-ITM shifts the learning target to GM, which is applied in the display process, thereby achieving a more flexible up-conversion than traditional ITM tasks.

## 3 FORMULATION

In this section, we elaborate on the pipeline of restoring linear HDR based on the SDR-GM pair. The detailed process is shown in Fig. 2, which can be divided into the following three steps:

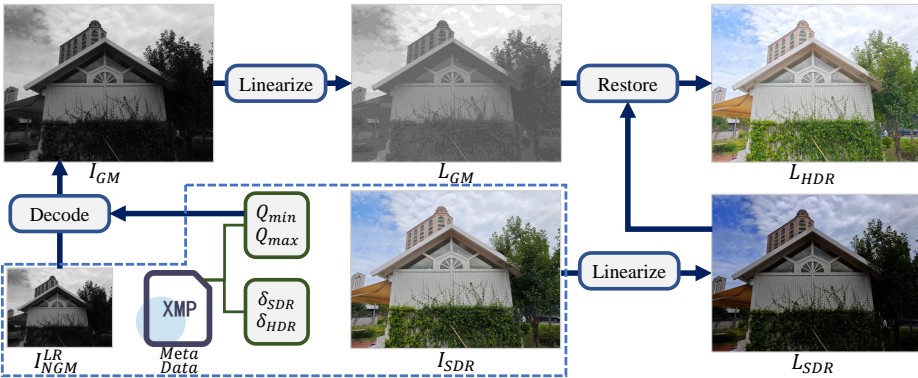

Figure 2: The SDR-GM pair to linear HDR pipeline. It processes an SDR image $I_{SDR}$, the down-sampled and normalized GM $I_{NGM}^{LR}$, and metadata (shown in the blue frame), to produce a linear HDR output for HDR display. The metadata includes the maximum value $Q_{max}$ and minimum value $Q_{min}$ for normalization, as well as the offsets $\delta_{SDR}$ and $\delta_{HDR}$ ensuring non-zero division when computing GM. The path of offsets in the figure is omitted for brevity.

(1) **Decode GM.** Practically, the GM, denoted as $I_{GM}$, is produced by pixel-wise division of HDR and SDR, followed by logarithmic compression, and finally normalization and down-sampling. The maximum and minimum values for normalization, $Q_{max}$ and $Q_{min}$, are stored in the metadata. To decode $I_{GM}$ from the file, we first apply the following equation:

$$I_{GM} = \mathrm{U}(I_{NGM}^{LR}) \times (Q_{max} - Q_{min}) + Q_{min}, \tag{1}$$

where $I_{NGM}^{LR}$ denotes the stored normalized and down-sampled GM, and $\mathrm{U}(\cdot)$ represents up-sampling function. In practice, the encoding and decoding process usually sets $Q_{min}$ as zero.

(2) **Linearize SDR and GM.** To obtain the final linear HDR, we need to linearize both the SDR image and GM. To be specific, $I_{SDR}$ is linearized to $L_{SDR}$ by electro-optical transfer function (EOTF), while $I_{GM}$ is linearized to $L_{GM}$ by an exponential function. The process can be formulated as:

$$\begin{aligned} L_{SDR} &= \mathrm{EOTF}(I_{SDR}), \\ L_{GM} &= \exp2(I_{GM}), \end{aligned} \tag{2}$$

where $\mathrm{EOTF}(\cdot)$ denotes EOTF function, and $\exp2(\cdot)$ denotes the exponential function with base 2.

(3) **Restore HDR.** Once the GM is decoded and both the SDR image and the GM are linearized, the linear HDR $L_{HDR}$ can be computed as follows:

$$L_{HDR} = (L_{SDR} + \delta_{SDR}) \odot L_{GM} - \delta_{HDR}, \tag{3}$$

where $\odot$ denotes element-wise multiplication, $\delta_{SDR}$ and $\delta_{HDR}$ are small offsets to prevent zero division in the encoding process. In practice, the Eq. 3 can also be simplified to $L_{HDR} = L_{SDR} \odot L_{GM}$ shown in Fig. 1 (a). More details of the simplification process and GM formation pipeline can be found in Sec. A of the appendix.

## 4 METHOD

### 4.1 OVERVIEW

As an intermediary in the ITM process, GM is characterized by high contrast along with fluctuating peak value, which makes direct estimation challenging. To tackle this challenge, we turn to the decomposition of GM, *i.e.*, the normalized GM $I_{NGM}$ and the corresponding maximum value $Q_{max}$. This decomposition is well-suited to the inherent locality and globality characteristics of GM, resulting in a more stable and robust learning process. Building on this, we propose a dual-branch network named GMNet, which consists of a Local Contrast Restoration (LCR) branch estimating $I_{NGM}$ from the pixel-level image features, and a Global Luminance Estimation (GLE) branch predicting $Q_{max}$ based on the image-level luminance information.

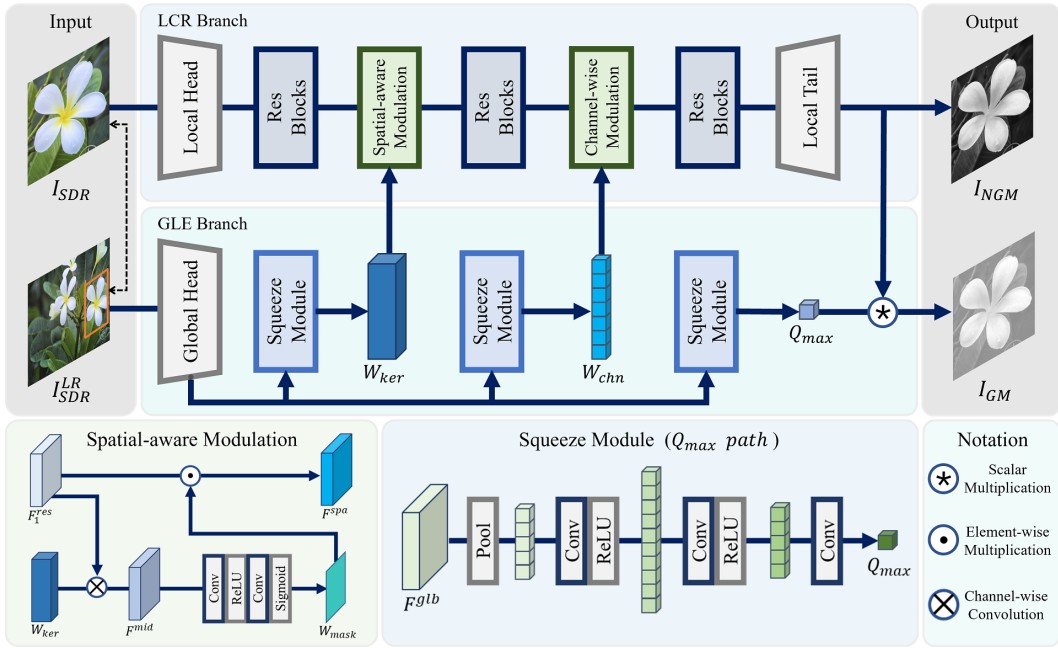

Figure 3: Architecture of the GMNet. The proposed network consists of a Local Contrast Restoration (LCR) branch, which predicts the normalized GM $I_{NGM}$, and a Global Luminance Estimation (GLE) branch, which estimates the maximum value $Q_{max}$ and provides global guidance to the LCR branch. The details of the spatial-aware modulation and the squeeze module are illustrated at the bottom. The final $I_{GM}$ is derived from the scalar multiplication of estimated $I_{NGM}$ and $Q_{max}$. During the training phase, the LCR branch processes the SDR image patch, while in inference phase, it handles the full-resolution SDR image. The GLE branch processes the down-sampled full SDR image in both the training and inference phases.

As depicted in Fig. 3, the LCR branch processes the SDR image $I_{SDR}$, extracts image features, and applies modulation in both spatial and channel dimensions, ending up with the normalized GM $I_{NGM}$ as output. The GLE branch takes the down-sampled SDR image $I_{SDR}^{LR}$ as input to reduce computational complexity, expands the receptive field, and further squeezes the feature to derive the maximum value $Q_{max}$. In addition to estimating $Q_{max}$, the intermediate features in the GLE branch also serve as the global guidance to enhance the restoration in the LCR branch. Specifically, the GLE branch extracts the spatial-aware modulation kernel $W_{ker}$ for spatial feature enhancement, and generates the channel-wise modulation weights $W_{chn}$ to modulate the regional features along the channel dimension. Finally, $I_{GM}$ is obtained by the multiplication of $I_{NGM}$ and $Q_{max}$, then following the pipeline in Sec. 3 to restore the final HDR result.

## 4.2 LOCAL CONTRAST RESTORATION BRANCH

To promote the restoration of the regional contrast, we introduce the LCR branch. It not only extracts fine-grained spatial details but also integrates spatial-aware and channel-wise modulation, which incorporate global guidance to refine and adjust the local prediction process, ensuring more accurate regional restoration.

As illustrated in Fig. 3, the input SDR image $I_{SDR}$ is initially processed into a shallow spatial feature $F^{loc}$ through a local head consisting of three convolutional layers with ReLU activation functions. The stride of the first convolution layer is set to 2 for a larger receptive field, facilitating subsequent convolutions to capture more information. The resulting $F^{loc}$ is then passed through three cascaded ResBlock groups (He et al., 2016) to extract deeper image features. Spatial-aware and channel-wise modulation are applied between the groups to effectively incorporate global information. Finally, a local tail with a pixel-shuffle layer and three convolutional layers reconstructs the feature into the normalized GM which is denoted as $I_{NGM}$.

**Spatial-aware modulation and channel-wise modulation.** During the spatial-aware modulation process, the LCR branch receives a spatial kernel $W_{ker} \in \mathbb{R}^{3 \times 3 \times C}$ from the GLE branch, which serves as global guidance to promote contrast restoration. The output feature $F_1^{res}$ of the first Res-Block group is first passed through a channel-wise convolution layer with spatial kernel $W_{ker}$, producing an intermediate feature $F^{mid}$. This intermediate feature is then processed through two consecutive $3 \times 3$ convolutional layers with a ReLU activation in the interval. The resulting feature is passed through the Sigmoid function to generate the spatial mask $W_{mask}$, which is applied to $F_1^{res}$ via element-wise multiplication, outputting the modulated spatial features $F^{spa}$ accordingly.

During the channel-wise modulation process, the LCR branch receives channel weights $W_{chn} \in \mathbb{R}^{1 \times 1 \times C}$ from the GLE branch, which are used to modulate the intermediate output $F_2^{res}$ of the second ResBlock group through a channel attention mechanism. These proposed modulation mechanisms enable effective global guidance in the local restoration process, making regional GM estimation more precise.

### 4.3 GLOBAL LUMINANCE ESTIMATION BRANCH

To facilitate the estimation of the fluctuating peak value of GM and provide global modulation guidance to the local contrast restoration process, we introduce the GLE branch to extract global features from the down-sampled SDR image $I_{SDR}^{LR}$. The proposed GLE branch not only produces an estimation on the peak value of the target GM, but also provides the spatial kernel $W_{ker}$ and channel weights $W_{chn}$. Specifically, the input $I_{SDR}^{LR}$ is first convolved to $F^{glb}$ by the global head, which shares a similar structure with the local head but conducts down-sampling twice. Next, $F^{glb}$ is fed into three parallel squeeze modules to derive the spatial kernel $W_{ker}$, channel weights $W_{chn}$ and the estimated peak value $Q_{max}$ required for reconstruction.

**Squeeze module.** To squeeze the global feature of image statistics, the squeeze module first down-samples $F^{glb}$ through an adaptive pooling layer, which is then expanded to richer feature space by two $1 \times 1$ convolutions and activation functions to accommodate more complex nonlinear mappings. Finally, a convolutional layer with $1 \times 1$ filters is used to adjust the shape of the final output.

To meet the different needs of guidance and estimation, the squeeze module controls the shape of the output through the pooling size of the first layer and the number of filters in the last layer. Specifically, the $3 \times 3$ pooling and $C$ filters in the last layer produce $W_{ker} \in \mathbb{R}^{3 \times 3 \times C}$, while $1 \times 1$ pooling and $C$ filters are employed to obtain $W_{chn} \in \mathbb{R}^{1 \times 1 \times C}$, and $Q_{max} \in \mathbb{R}$ is derived from $1 \times 1$ pooling and convolution with one filter at last.

### 4.4 LOSS FUNCTION

Directly supervising $I_{NGM}$ and $Q_{max}$ can lead to imbalanced supervision between a tensor and a scalar. To address this, we retain direct supervision to $I_{NGM}$ and indirectly supervise $Q_{max}$ through $I_{GM}$, which is the product of $I_{NGM}$ and $Q_{max}$. The overall loss function in the proposed framework consists of two components. The first is the $\mathcal{L}_1$ loss for $I_{NGM}$, defined as follows:

$$\mathcal{L}_{NGM} = \left\| I_{NGM} - \hat{I}_{NGM} \right\|_1, \tag{4}$$

where $\hat{I}_{NGM}$ denotes the estimated normalized GM. The second component is the $\mathcal{L}_1$ loss for $I_{GM}$, which is defined as follows:

$$\mathcal{L}_{GM} = \left\| I_{GM} - \hat{I}_{GM} \right\|_1, \tag{5}$$

where $\hat{I}_{GM}$ denotes the calculated GM from the Eq. 1. To ensure uniform scaling and smooth convergence, $I_{GM}$ is normalized by the dataset maximum for supervision. In summary, the total loss is defined as:

$$\mathcal{L}_{total} = \alpha_1 \mathcal{L}_{NGM} + \alpha_2 \mathcal{L}_{GM}, \tag{6}$$

where $\alpha_1$ and $\alpha_2$ are weights used to balance the different components of the loss function. In our implementation, we set $\alpha_1 = 1.0$ and $\alpha_2 = 3.0$.

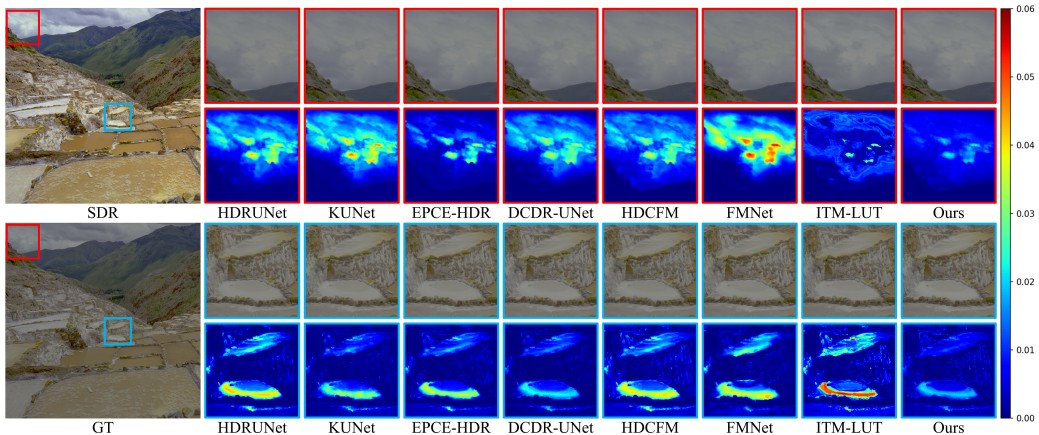

Figure 4: Qualitative comparisons on the synthetic dataset. The left side shows the SDR image and ground truth PQ-encoded HDR image, while the right side displays the output patches and corresponding error maps.

| Category | Method | Params | Linear Domain | | | PQ Domain | | | HDR Domain | |
|---|---|---|---|---|---|---|---|---|---|---|
| | | | PSNR↑ | SSIM↑ | SRSIM↑ | PSNR↑ | SSIM↑ | SRSIM↑ | $\Delta E_{ITP}$↓ | HDR-VDP3↑ |
| SI-HDR | HDRUNet | 1.57M | 41.2579 | 0.9969 | 0.9950 | 40.9315 | 0.9946 | 0.9988 | 3.5799 | 9.9080 |
| | KUNet | 1.08M | 40.4879 | 0.9952 | 0.9943 | 34.5201 | 0.9619 | 0.9986 | 7.1538 | 9.7851 |
| | EPCE-HDR | 33.79M | 41.5919 | 0.9956 | 0.9954 | 31.4700 | 0.9430 | 0.9916 | 8.5830 | 9.6694 |
| | DCDR-UNet | 1.83M | 41.8085 | 0.9971 | 0.9952 | 42.0183 | 0.9962 | 0.9988 | 3.0501 | 9.9215 |
| SDR-to-HDRTV | HDCFM | 0.10M | 41.8563 | 0.9967 | 0.9960 | 44.9017 | 0.9992 | 0.9995 | 2.4773 | 9.9115 |
| | FMNet | 1.24M | 40.7699 | 0.9970 | 0.9940 | 44.4798 | 0.9996 | 0.9994 | 2.3032 | 9.8991 |
| | ITM-LUT | 0.57M | 39.5735 | 0.9950 | 0.9912 | 43.7152 | 0.9987 | 0.9987 | 2.2865 | 9.7268 |
| GM-ITM | Ours | 1.83M | **43.5510** | **0.9977** | **0.9976** | **47.6256** | **0.9998** | **0.9997** | **1.6262** | **9.9477** |

Table 2: Quantitative comparisons on the synthetic dataset. PSNR, SSIM, and SRSIM are used to evaluate both linear HDR and PQ-encoded HDR. Additionally, $\Delta E_{ITP}$ and HDR-VDP3, metrics specifically designed for HDR evaluation, are employed. **Bold** text and underline text indicate the best and second-best performance, respectively.

## 5 EXPERIMENTS

### 5.1 IMPLEMENTATION DETAILS

We use the Adam optimizer (Kingma, 2014) with $\beta_1 = 0.9$ and $\beta_2 = 0.99$ to train our network. The batch size is set to 32, and the initial learning rate is $2 \times 10^{-4}$, halving every $2 \times 10^4$ iterations, with a total of $1 \times 10^5$ iterations. The weights of GMNet are randomly initialized. Each ResBlock group contains 5 blocks, and the number of hidden layers $C$ is set to 64. The input of the GLE branch $I_{SDR}^{LR}$ is at the reduced $256 \times 256$ resolution. We use the ReLU activation function in the network. All experiments are conducted on a workstation equipped with RTX 3090 under Ubuntu 20.04 LTS.

### 5.2 DATASETS

**Synthetic dataset.** We collect eleven BT.2020/PQ 4K-UHD HDR videos from the internet, using eight for training and three for testing. To ensure diversity and reduce redundancy, frames are extracted at two-second intervals, resulting in a total of 1276 frames. Corresponding SDR images are generated using our degradation model, which incorporates roll-off, gamut mapping, tone mapping, clipping, and quantization (Reinhard et al., 2023; ITU-R, 2017; 2021; 2023). The GM is then calculated by Eq. 8, where $Q_{min}, \delta_{min}, \delta_{max}$ are set to zero. Both the GM and SDR have a resolution of $3840 \times 2160$. After filtering, we obtain a training set of 900 pairs and a test set of 100 pairs.

**Real-world dataset.** To further explore the complexity of the real-world imaging environment, we build a real-world dataset to advance research in GM-ITM. We capture images in various natural settings using the Xiaomi 14 Ultra equipped with the LYT-900 sensor, which outputs double-layer HDR images in ISO standard (ISO, 2024), and $Q_{min}, \delta_{min}, \delta_{max}$ are zero by default. The dataset

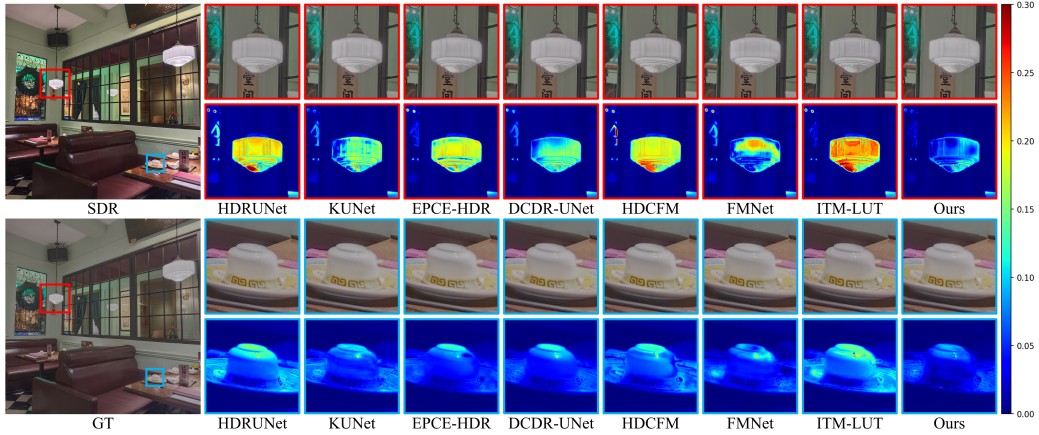

Figure 5: Qualitative comparisons on the real-world dataset. The left side shows the SDR image and ground truth PQ-encoded HDR image, while the right side displays the output patches and corresponding error maps.

| Category | Method | Params | Linear Domain | | | PQ Domain | | | HDR Domain | |
|---|---|---|---|---|---|---|---|---|---|---|
| | | | PSNR↑ | SSIM↑ | SRSIM↑ | PSNR↑ | SSIM↑ | SRSIM↑ | $\Delta E_{ITP}$↓ | HDR-VDP3↑ |
| SI-HDR | HDRUNet | 1.57M | 33.1686 | 0.9911 | 0.9866 | 37.3966 | 0.9969 | 0.9984 | 5.7358 | 9.8455 |
| | KUNet | 1.08M | 33.3256 | 0.9907 | 0.9880 | 33.7895 | 0.9690 | 0.9979 | 9.0578 | 9.6893 |
| | EPCE-HDR | 33.79M | 32.7616 | 0.9914 | 0.9859 | 31.1207 | 0.9466 | 0.9973 | 10.4322 | 9.7218 |
| | DCDR-UNet | 1.83M | 33.5421 | 0.9924 | 0.9873 | 38.3252 | 0.9970 | 0.9987 | 4.8645 | **9.8781** |
| SDR-to-HDRTV | HDCFM | 0.10M | 32.7963 | 0.9879 | 0.9882 | 38.1894 | 0.9956 | 0.9984 | 6.2957 | 9.8035 |
| | FMNet | 1.24M | 32.6247 | 0.9921 | 0.9828 | 39.7632 | 0.9988 | 0.9986 | 4.5544 | 9.8110 |
| | ITM-LUT | 0.57M | 31.6677 | 0.9842 | 0.9871 | 37.9021 | 0.9958 | 0.9985 | 5.4299 | 9.6831 |
| GM-ITM | Ours | 1.83M | **33.9490** | **0.9928** | **0.9896** | **40.2088** | **0.9993** | **0.9989** | **4.0260** | 9.8757 |

Table 3: Quantitative comparisons on the real-world dataset. PSNR, SSIM, and SRSIM are used to evaluate both linear HDR and PQ-encoded HDR. Additionally, $\Delta E_{ITP}$ and HDR-VDP3, metrics specifically designed for HDR evaluation, are employed. **Bold** text and underline text indicate the best and second-best performance, respectively.

includes diverse scenes, both indoor and outdoor, as well as daytime and nighttime settings, covering a wide range of brightness levels within some scenes. The SDR image has a resolution of $4096 \times 3072$, while the GM is $2048 \times 1536$, and both horizontal and vertical orientations are included. After filtering out low-quality pairs, we select 900 pairs for training and 100 pairs for testing. More details of the datasets can be found in Sec. B of the appendix.

## 5.3 COMPARISONS TO OTHER METHODS

**Evaluation metrics.** To comprehensively validate the performance of HDR-related methods on GM-ITM, we quantitatively compare HDR results across three domains. In the linear domain, where HDR results retain their original linear form with highly varying peak values, directly computing metrics may lead to unbalanced comparisons. To address this, we evaluate the HDR results in their normalized form using PSNR, SSIM, and SRSIM (Zhang & Li, 2012) metrics. In the PQ domain, HDR results are encoded with the PQ function, which offers a more perceptually reflective measure of visual similarity. Similarly, we normalize the PQ-encoded HDR and employ the same metrics used in the linear domain. Additionally, we use HDR-specific metrics such as $\Delta E_{ITP}$ (ITU-R, 2019) and HDR-VDP3 (Mantiuk et al., 2023), which we collectively refer to as the HDR domain.

**Compared methods.** To the best of our knowledge, there are currently no methods specifically designed for GM-ITM. Therefore, we compare our method with the solutions of closely related tasks, *i.e.*, SI-HDR, and SDR-to-HDRTV up-conversion. For SI-HDR, where the learning target is to restore linear HDR, we use HDRUNet (Chen et al., 2021a), KUNet (Wang et al., 2022), EPCE-HDR (Tang et al., 2023), and DCDR-UNet (Kim et al., 2024) for comparison. For SDR-to-HDRTV up-conversion, in which the learning target is to restore the PQ-encoded HDR images, we compare our method with HDCFM (He et al., 2022), FMNet (Xu et al., 2022), and ITM-LUT (Guo et al., 2023b). All methods are re-trained on our proposed datasets to ensure a fair comparison.

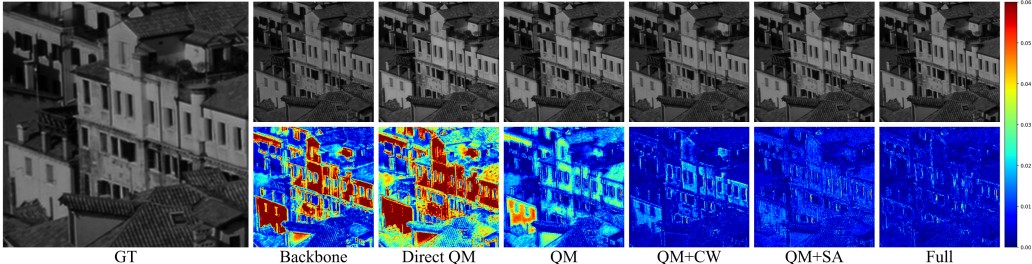

Figure 6: The qualitative ablation results of the proposed components. The corresponding error maps are shown below the normalized GM estimation results.

| Backbone | Components | | | Evaluation Metrics | | | | |
|---|---|---|---|---|---|---|---|---|
| | QM | CW | SA | PSNR-L↑ | PSNR-PQ↑ | PSNR-NGM↑ | $\Delta E_{ITP}$↓ | HDR-VDP3↑ |
| ✓ | - | - | - | 41.1670 | 45.2765 | 29.5436 | 1.7711 | 9.9210 |
| ✓ | ◯ | - | - | 41.3755 | 44.9717 | 28.4058 | 2.1757 | 9.9128 |
| ✓ | ✓ | - | - | 42.3479 | 46.2067 | 32.8009 | 1.8395 | 9.9431 |
| ✓ | ✓ | ✓ | - | 43.0480 | 47.1238 | 33.2117 | 1.6939 | 9.9454 |
| ✓ | ✓ | - | ✓ | 42.9085 | 47.0011 | 33.2731 | 1.7081 | 9.9442 |
| ✓ | ✓ | ✓ | ✓ | 43.5510 | 47.6256 | 33.4729 | 1.6262 | 9.9477 |

Table 4: The quantitative ablation results of the proposed components. The QM, CW, and SA represent $Q_{max}$ prediction, channel-wise modulation, and spatial-aware modulation. The "-", "◯", and "✓" in the QM column represent not, directly, and indirectly learning $Q_{max}$.

**Quantitative comparison.** Tab. 2 lists the quantitative results on the synthetic dataset. Through analysis, we find that the SI-HDR methods achieve better quantitative results in the linear domain, while inferior to the SDR-to-HDRTV methods in the PQ domain. It is easy to understand that different optimization targets lead to bias in two domains. An exception is the HDCFM, which efficiently fits the synthetic dataset with lightweight parameters and avoids over-fitting, achieving good results in the linear domain. The target of our method is neither linear HDR nor PQ-encoded HDR, but the best results are achieved in both domains, proving the superiority of targeting GM and the effectiveness of the proposed method.

The quantitative results on the real-world dataset are listed in Tab. 3. Compared to synthetic data, real-world data is derived from a black-box pipeline, features a higher level of noise, and involves more complex imaging environments. Our method still achieves the best results for complex real-world data, proving the generalization and effectiveness of our method. More details of comparison experiments on computational efficiency and real-time processing capability can be found in Sec. C of the appendix.

**Qualitative comparison.** As the high dynamic range display is not always available, we follow previous work (Chen et al., 2021b; Xu et al., 2022), encoding the linear HDR results with PQ-OETF for visualization. Since the SDR display is usually decoded by Gamma-EOTF, the images may appear dark here. In addition, we compute the error maps between the predicted HDR and the GT in the linear domain, which are shown below the patches.

The visualization results of the synthetic dataset and the real dataset are shown in Fig. 4 and Fig. 5, and more visual comparisons can be found in Sec. D, Sec. E, and Sec. F of the appendix. For the SI-HDR methods, the visual results tend towards significant differences over small areas and distortions in details. The SDR-to-HDRTV methods are short in the highlight pixels due to the over-compression of the PQ-OETF, and also perform poorly in contrast restoration. Our method can effectively reconstruct the overall luminance and regional contrast details, especially the highlight details, achieving a more realistic and compelling visual experience.

## 5.4 ABLATION STUDIES

**Ablation on the proposed components.** We conduct an ablation analysis on three main components of GMNet in the proposed synthetic dataset, and the quantitative results are shown in Tab. 4.

| Hidden Layers | Params | Linear Domain | | | PQ Domain | | | HDR Domain | |
|---|---|---|---|---|---|---|---|---|---|
| | | PSNR↑ | SSIM↑ | SRSIM↑ | PSNR↑ | SSIM↑ | SRSIM↑ | $\Delta E_{ITP}\downarrow$ | HDR-VDP3↑ |
| 32 | 0.46M | 42.7029 | 0.9972 | 0.9972 | 46.7620 | 0.9997 | 0.9997 | 1.7093 | 9.9319 |
| 48 | 1.03M | 43.3299 | 0.9973 | 0.9973 | 47.4524 | 0.9998 | 0.9997 | 1.6990 | 9.9408 |
| 64 | 1.83M | 43.5510 | 0.9977 | 0.9976 | 47.6256 | 0.9998 | 0.9997 | 1.6262 | 9.9477 |

Table 5: Ablation study on the model size of GMNet. The original implementation of our network is 64 hidden layers.

| Down Scale | Model | Linear Domain | | | PQ Domain | | | HDR Domain | |
|---|---|---|---|---|---|---|---|---|---|
| | | PSNR↑ | SSIM↑ | SRSIM↑ | PSNR↑ | SSIM↑ | SRSIM↑ | $\Delta E_{ITP}\downarrow$ | HDR-VDP3↑ |
| ×1 | HDRUNet | 41.2579 | 0.9969 | 0.9950 | 40.9315 | 0.9946 | 0.9988 | 3.5799 | 9.9080 |
| | HDCFM | 41.8563 | 0.9967 | 0.9960 | 44.9017 | 0.9992 | 0.9995 | 2.4773 | 9.9115 |
| | Ours | 43.5510 | 0.9977 | 0.9976 | 47.6256 | 0.9998 | 0.9997 | 1.6262 | 9.9477 |
| ×2 | HDRUNet | 39.3861 | 0.9969 | 0.9952 | 40.6861 | 0.9969 | 0.9993 | 2.7233 | 9.9038 |
| | HDCFM | 39.6308 | 0.9964 | 0.9953 | 44.4200 | 0.9993 | 0.9995 | 2.6179 | 9.9178 |
| | Ours | 41.2254 | 0.9974 | 0.9972 | 46.5649 | 0.9998 | 0.9997 | 1.7124 | 9.9461 |
| ×4 | HDRUNet | 38.9650 | 0.9970 | 0.9951 | 39.9763 | 0.9962 | 0.9992 | 2.7883 | 9.9046 |
| | HDCFM | 39.2906 | 0.9966 | 0.9961 | 44.3981 | 0.9993 | 0.9995 | 2.4795 | 9.9178 |
| | Ours | 40.5103 | 0.9972 | 0.9966 | 46.4938 | 0.9998 | 0.9996 | 1.7672 | 9.9413 |

Table 6: Ablation study on the resolution of GM. We compare the results of HDRUNet (Chen et al., 2021a), HDCFM (He et al., 2022), and our method in three different down-sampling scales.

Compared to not learning $Q_{max}$, learning $Q_{max}$ directly gets better linear performance, but the imbalanced supervision between tensor and scalar causes a decline in perception metrics, and all the results have a significant decay compared to indirectly learning $Q_{max}$. Respectively introducing the spatial-aware modulation or channel-wise modulation improves the network performance, and applying both makes the full net obtain the highest quantitative results.

We also visualize the predicted normalized GM for the ablation study, and the results are shown in Fig. 6. The experiment without $Q_{max}$ path obtains poor estimation both locally and globally, while achieving significant improvement by attaching it. The sole application of spatial-aware modulation or channel-wise modulation makes a modest increment, and the application of both achieves further visual improvement. The results of both quantitative and qualitative ablation experiments validate the effectiveness of the proposed components.

**Ablation on the model size of GMNet.** Keeping the network structure unchanged, we adjust the number of hidden layers to control the model size. The results of the ablation experiments on the synthetic dataset are shown in Tab. 5. The experimental results show that reducing model parameters does not lead to significant performance degradation, especially in SSIM-L, SSIM-PQ and HDR-VDP3 that measure perception quality, verifying that our method can maintain accuracy with reduced resources.

**Ablation on the resolution of GM.** As mentioned in Sec. 3, GM is usually down-sampled to reduce file size. Therefore, we conduct ablation experiments on three different down-scales of GM on the synthetic dataset, analyzing the effect of resolution on up-conversion quality as shown in Tab. 6. For different down-sampling factors, our method is kept in a leading position. Though the increment slightly decays with the larger multiples, our method requires less computation while the other methods remain the same, proving the effectiveness of our method.

# 6 CONCLUSION

Motivated by a novel double-layer HDR image format, we introduce a Gain Map-based Inverse Tone Mapping (GM-ITM) task, for which we specially design a network named GMNet. Based on the analysis of GM characteristics, we introduce a Local Contrast Restoration (LCR) branch and a Global Luminance Estimation (GLE) branch for GMNet, which capture pixel-wise and image-wise information for GM estimation. Moreover, we first present synthetic and real-world datasets consisting of SDR-GM pairs to promote the research of GM-ITM. Experimental results demonstrate the superiority of our method in qualitative and quantitative evaluations.

ACKNOWLEDGMENTS

This work was supported in part by the National Natural Science Foundation of China under Grants 62131003 and 62021001.

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

## A  GM FORMATION PIPELINE

Corresponding to the SDR-GM pair to HDR pipeline introduced in Sec. 3, we illustrate how to calculate GM from the SDR-HDR pair in this section. First, the GM is computed by the following equation:

$$I_{GM} = log_2 \left( \frac{L_{HDR} + \delta_{HDR}}{L_{SDR} + \delta_{SDR}} \right), \tag{7}$$

where a common empirical value for $\delta_{SDR}$ and $\delta_{HDR}$ is $1/64$. The Eq. 7 can also be conducted as follows:

$$I_{GM} = \begin{cases} log_2 \left( \frac{L_{HDR}}{L_{SDR}} \right) & , L_{SDR} \neq 0, \\ 0 & , L_{SDR} = 0, \end{cases} \tag{8}$$

which is free of offsets $\delta_{SDR}$ and $\delta_{HDR}$, allowing Eq. 3 to be simplified to:

$$L_{HDR} = L_{SDR} \odot L_{GM}. \tag{9}$$

To reduce file size, the calculated $I_{GM}$ is then normalized using the following equation:

$$I_{NGM} = \frac{I_{GM} - Q_{min}}{Q_{max} - Q_{min}}. \tag{10}$$

Finally, the $I_{NGM}$ will be down-sampled to $I_{NGM}^{LR}$ for further compression, and $I_{NGM}^{LR}$ may be gamma-encoded in some cases.

In practice, many implements (Apple, 2021; Google, 2024) recommend using the linear luminance $Y_{SDR}$ and $Y_{HDR}$ to substitute $L_{SDR}$ and $L_{HDR}$ in Eq. 7 or Eq. 8, thus making $I_{GM}$ a gray-scale GM. To our knowledge, the existing devices can only produce gray-scale GM. Therefore, all experiments in this paper are performed on the single-channel GM, and the three-channel GM is beyond our scope.

## B  DETAILS OF DATASETS

We investigate the datasets employed in SI-HDR and SDR-to-HDRTV up-conversion tasks, and compare them with our proposed synthetic and real-world dataset in Tab. 8.

Besides, we demonstrate the diversity of the synthetic dataset and real-world dataset in Fig. 7 and Fig. 8, which includes outdoor, indoor, daytime, and nighttime scenes, providing a wide range of content with different brightness levels and texture types.

## C  REAL-TIME PERFORMANCE EVALUATION

To evaluate the computational efficiency and real-time processing capability of our method, we conduct comparison experiments in Tab. 7.

| Method | Params↓ | Runtime↓ | MACs↓ | PSNR-L↑ | PSNR-PQ↑ | $\Delta E_{ITP}$↓ |
|--------|---------|----------|-------|---------|----------|------------|
| HDRUNet | 1.57M | 613.89ms | 4161.75G | 33.1686 | 37.3966 | 5.7358 |
| KUNet | 1.08M | 621.70ms | 7534.50G | 33.3256 | 33.7895 | 9.0578 |
| EPCE-HDR | 33.79M | 23992.84ms | 326412.14G | 32.7616 | 31.1207 | 10.4322 |
| DCDR-UNet | 1.83M | 1085.91ms | 4502.77G | 33.5421 | 38.3252 | 4.8645 |
| HDCFM | 0.10M | 497.05ms | 127.08G | 32.7963 | 38.1894 | 6.2957 |
| FMNet | 1.24M | 304.13ms | 4147.13G | 32.6247 | 39.7632 | 4.5544 |
| ITM-LUT | 0.57M | 19.81ms | 59.16G | 31.6677 | 37.9021 | 5.4299 |
| Ours | 1.83M | 75.26ms | 1112.18G | 33.9490 | 40.2088 | 4.0260 |

Table 7: Efficiency comparisons on the real-world dataset. The runtime is evaluated in NVIDIA A100 as the average of 100 trials in the resolution of $4096 \times 3072$.

As can be seen, benefiting from the simple form of the target GM, our method achieves the second fastest runtime. While the look-up-table-based solution ITM-LUT is faster, its performance on image quality metrics is much lower.

| Task | Dataset | Volume | Resolution | Input | | | Output | | |
|---|---|---|---|---|---|---|---|---|---|
| | | | | Type | Depth | Channel | Type | Depth | Channel |
| SI-HDR | Cordts et al. (2016) | 2975 images | 2048×1024 | LDR | 8bit | RGB | HDR | 16bit | RGB |
| | Pérez-Pellitero et al. (2021) | 1494 images | 1900×1060 | LDR | 8bit | RGB | HDR | 16bit | RGB |
| SDR-to-HDRTV | Kim et al. (2019) | 39840 patches | 160×160 | SDR | 8bit | YUV | HDR | 16bit | YUV |
| | Zeng et al. (2020) | 23229 patches | 1080×1080 | SDR | 8bit | YUV | HDR | 10bit | YUV |
| | Chen et al. (2021b) | 1235 images | 3840×2160 | SDR | 8bit | RGB | HDR | 16bit | RGB |
| | Guo et al. (2023a) | 3878 images | 3840×2160 | SDR | 8bit | RGB | HDR | 16bit | RGB |
| GM-ITM | synthetic dataset | 900 images | 3840×2160 | SDR | 8bit | RGB | GM | 8bit | GRAY |
| | real-world dataset | 900 images | 4096×3072 | SDR | 8bit | RGB | GM | 8bit | GRAY |

Table 8: Details on the training set of different HDR datasets. The image resolution of some datasets varies, thus we list the highest resolution of each dataset. Our datasets stand out for the first and second highest resolution and the unique GM output.

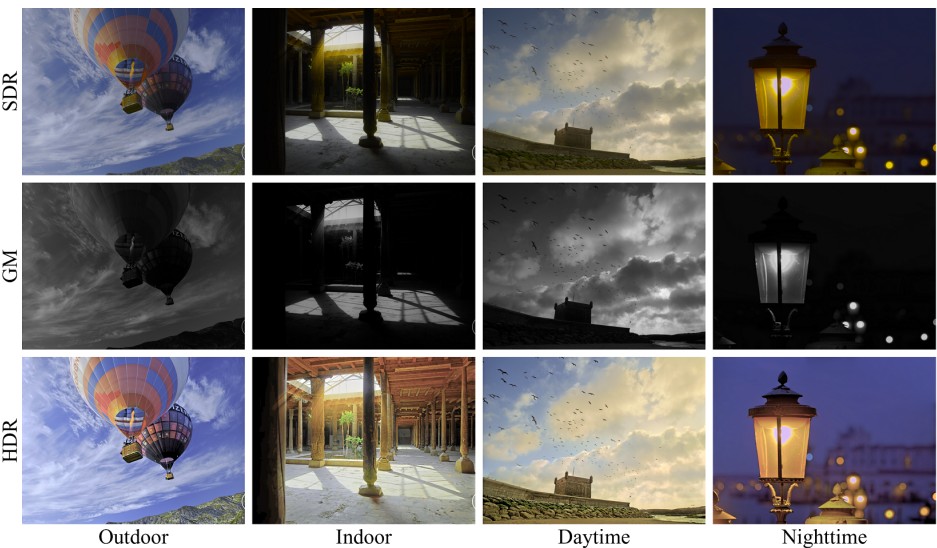

Figure 7: The diverse scenes, including outdoor, indoor, daytime, and nighttime settings, in the proposed synthetic dataset. The depicted HDR images are tone-mapped by the method proposed by Liang et al. (2018).

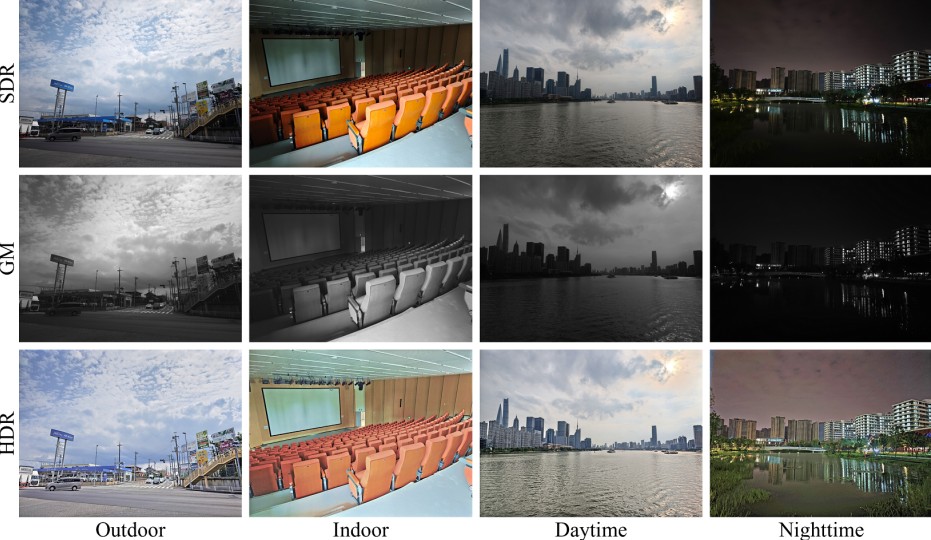

Figure 8: The diverse scenes, including outdoor, indoor, daytime, and nighttime settings, in the proposed real-world dataset. The depicted HDR images are tone-mapped by the method proposed by Liang et al. (2018).

## D ADDITIONAL VISUAL COMPARISONS ON DIVERSE SCENES

In this section, we provide more qualitative results for visual comparisons between our method and existing methods. For the results of the synthetic dataset shown in Fig. 9, our method reconstructs better edges than other methods, achieving the best visual quality in the highlights. We also visualize the estimation results on a real-world nighttime scene in Fig. 10. Our method stands out for the best perceptual quality and minimum errors, proving the effectiveness of our method in night scenes.

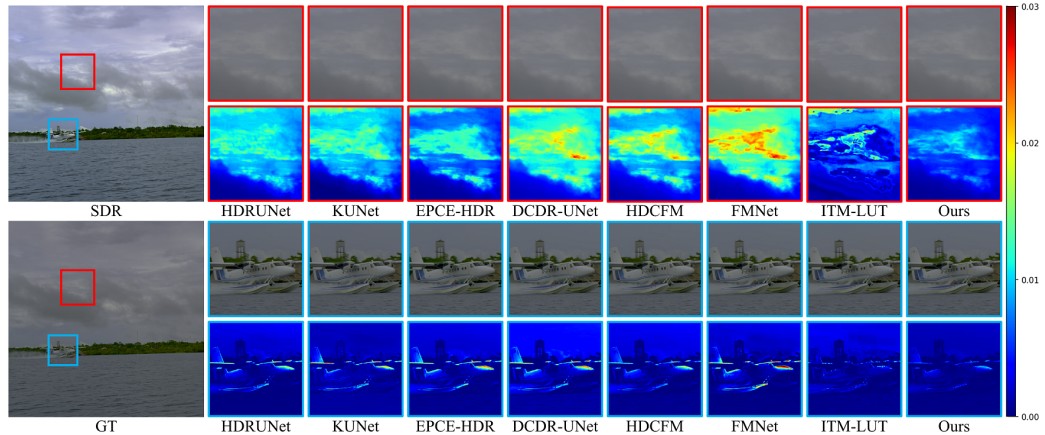

Figure 9: Qualitative comparisons on the synthetic dataset. The left side shows the SDR image and ground truth PQ-encoded HDR image, while the right side displays the output patches and corresponding error maps.

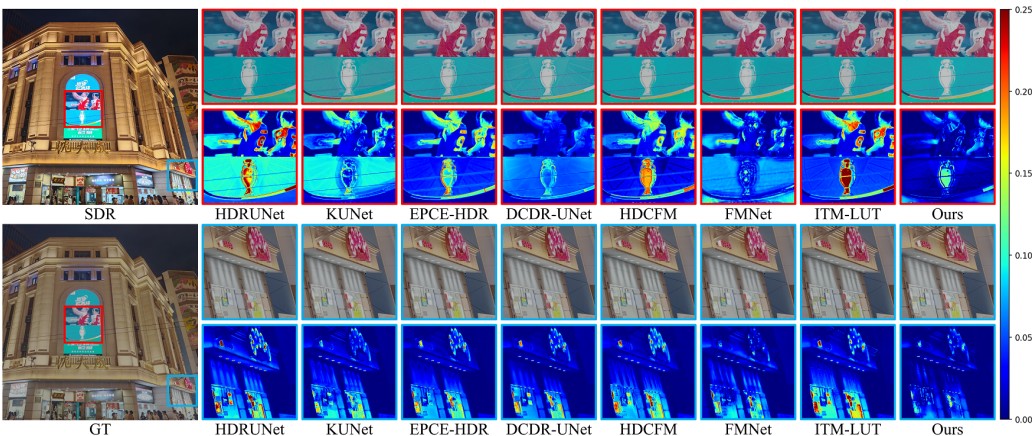

Figure 10: Qualitative comparisons on the real-world dataset. The left side shows the SDR image and ground truth PQ-encoded HDR image, while the right side displays the output patches and corresponding error maps.

## E ADDITIONAL VISUAL COMPARISONS ON CHALLENGING CONDITIONS

To enable more in-depth analysis in challenging conditions, we perform qualitative experiments in Fig. 11 and Fig. 12.

As shown in Fig. 11, our method achieves superior performance in the sun region at extreme brightness. The HDCFM also achieves low errors, but with grid effect due to operator properties. Fig. 12 shows the estimation results on the leaves with complex textures and eaves with sharp edges, and our method demonstrate superior performance on complex textures in the real-world scenario, validating the generalization and the robustness of the proposed method.

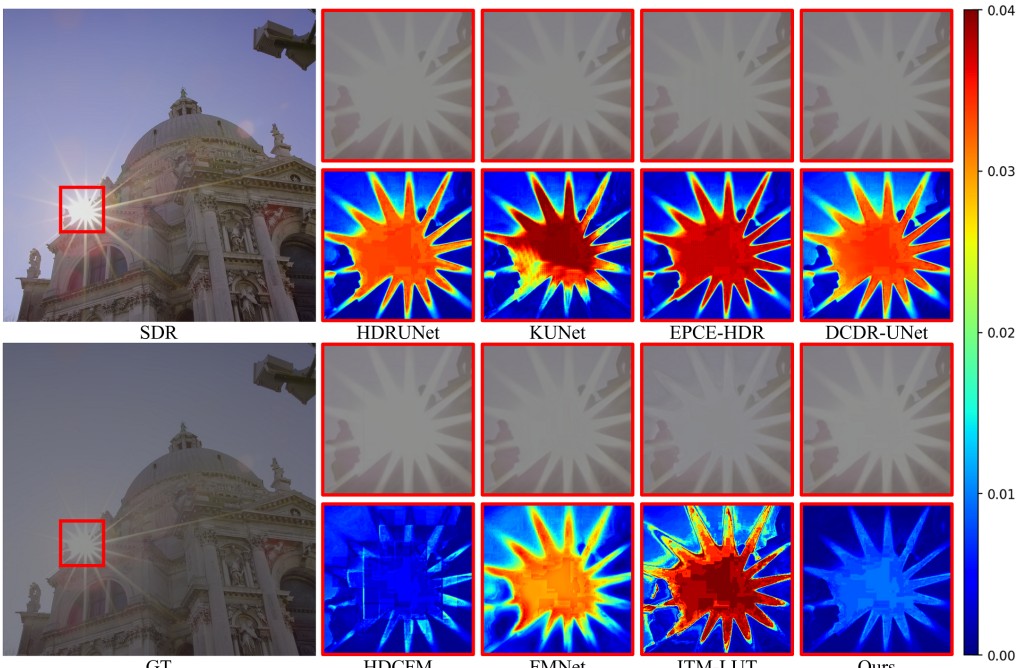

Figure 11: Qualitative comparisons on the synthetic dataset. The left side shows the SDR image and ground truth PQ-encoded HDR image, while the right side displays the output patches and corresponding error maps.

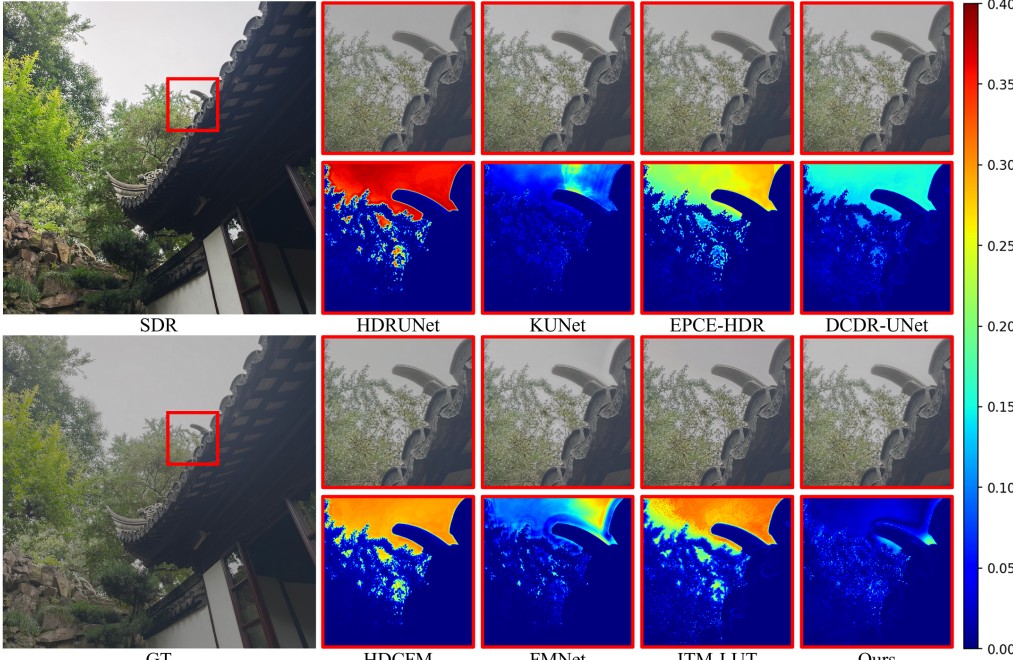

Figure 12: Qualitative comparisons on the real-world dataset. The left side shows the SDR image and ground truth PQ-encoded HDR image, while the right side displays the output patches and corresponding error maps.

## F    ADDITIONAL VISUAL COMPARISONS ON MORE DATASETS

The qualitative comparisons of diverse images from HDR-SYNTH (Liu et al., 2020), HDR-REAL (Liu et al., 2020), HDREye (Nemoto et al., 2015), and VDS (Lee et al., 2018b) datasets are meaningful and help to increase the credibility of our experiments. The results of the qualitative

experiments are shown in Fig. 13. Experimental results show that our method recovers local and global contrast well and achieves smooth and realistic visual results, verifying its generalizability over a wider range of data.

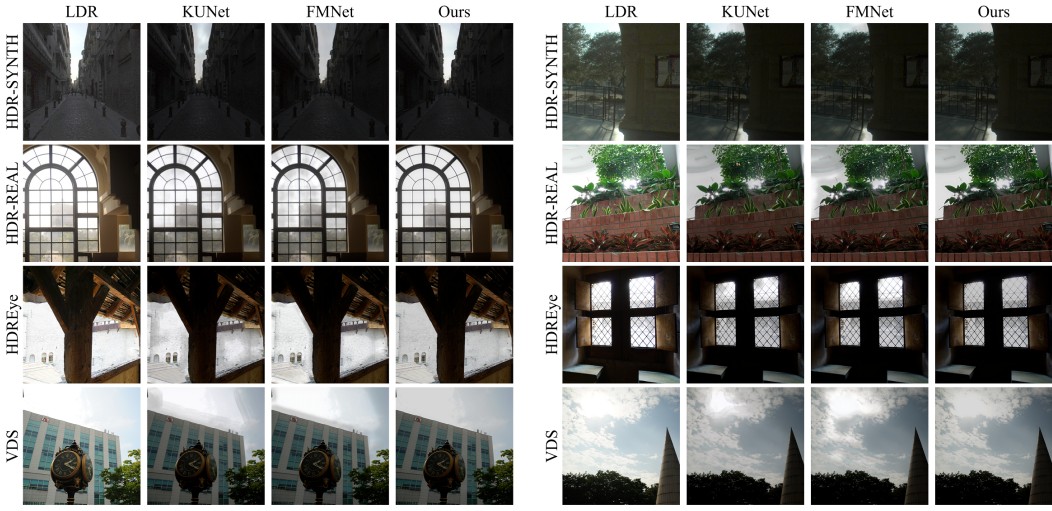

Figure 13: Qualitative comparisons on the HDR-SYNTH (Liu et al., 2020), HDR-REAL (Liu et al., 2020), HDREye (Nemoto et al., 2015), and VDS (Lee et al., 2018b) datasets.

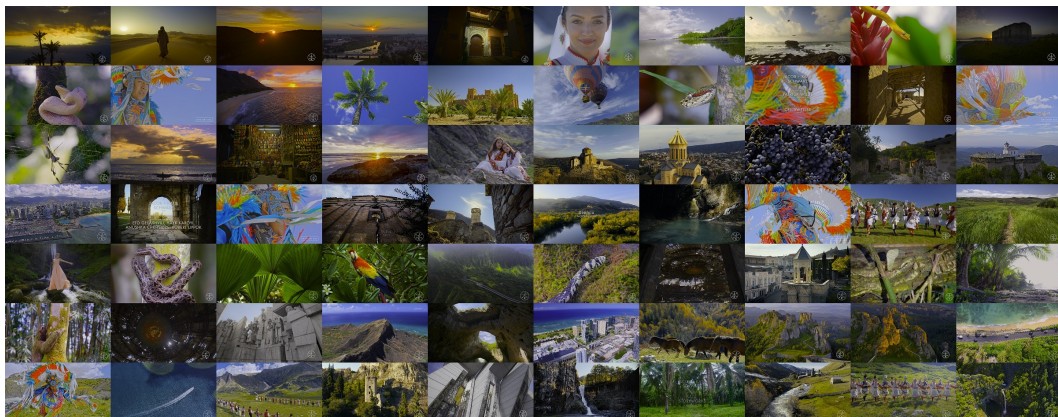

Figure 14: The thumbnails of the proposed synthetic dataset. It provides a wide range of content with different brightness levels and texture types.

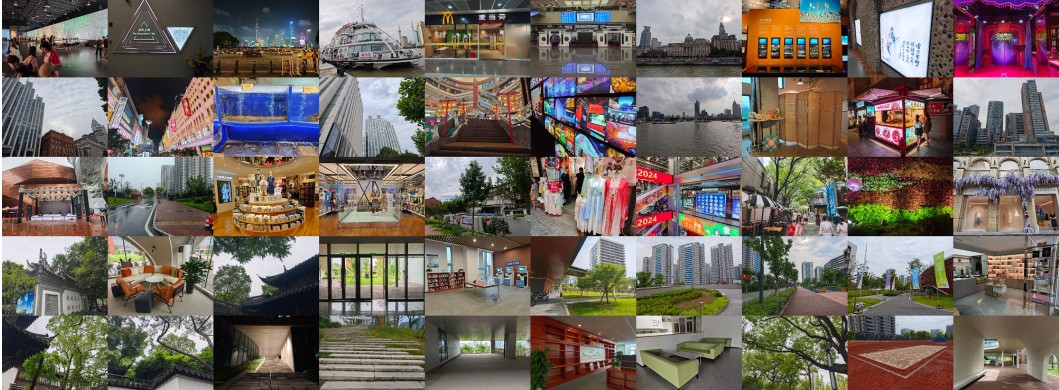

Figure 15: The thumbnails of the proposed real-world dataset. It provides a wide range of content with different brightness levels and texture types.

