# OpenReview forum: "Learning Gain Map for Inverse Tone Mapping"
_ICLR.cc/2025/Conference — ICLR 2025 Poster_

### Official Review · Reviewer_6tzz · 2024-11-01

**Soundness:** 2
**Presentation:** 2
**Contribution:** 2
**Rating:** 6
**Confidence:** 5

**Summary:**

The paper introduces a dual-branch network GMNet to estimate the Gain Map for Inverse Tone-mapping. The paper also builts a synthetic and a real-world dataset from the existing HDR resources and mobile devices, respectively.

**Strengths:**

The idea of estimating the Gain Map instead of the HDR value is interesting, which can become a baseline for any future research.

**Weaknesses:**

The methods that were used for comparison in the paper are outdated.
The method is not really novel.
Lack of a common metric in HDR domain.
There are also some technical issues about loss function, data and models.

**Questions:**

Although the idea is interesting, it is not new and quite similar to estimating the transfer function which is well-known in image restoration domain. That said the contribution of the paper is limited.

The literature reviews and compared methods are outdated which makes the paper less convincing. The latest method that was compared is from 2022 (KUNet) while the most recent one is DCDR-UNet: Deformable Convolution Based Detail Restoration via U-shape Network for Single Image HDR Reconstruction (based on Google Scholar).

In term of  PSNR in HDR domain, there is another metric which is more common than PNSR-NGM named mu-PSNR. The paper should consider running the evaluation on this metric too.

It is not clear about Q_max. Q_max value should not be in the range of [0,1], is that correct? If so, I_GM is not in the range [0,1] either. And can not normalize it as well, the reviewer wonder how the loss can be calculated and how the model was converged?

Finally, as there is a upsampling function when estimating I_GM, the reviewer think that that I_GM might not represent the real I_GM precisely due to the interpolation function when applying upsampling (which can be either nearest neighbors or bidirectional).

---

> ### Author Response · Authors · 2024-11-21
>
> We appreciate the reviewer's valuable comments and suggestions, and we hope our responses could address the concerns.
>
> **1. Similarity to the transfer function**
>
> We respectfully cannot agree. GM is NOT similar to the transfer function, which typically maps the input to the output with **a learned function** (implemented by neural networks nowadays). Suppose $f_{\phi}(\cdot)$ is a neural network, in the HDR task, it is implemented by $L_{HDR}=f_{\phi}(L_{SDR})$, where $\phi$ denotes the learned parameters of the network. In contrast, GM is an image-like auxiliary data that records pixel-wise dynamic range information, which is **independently collected along with the SDR image** and is implemented by $L_{HDR}=L_{GM}\odot L_{SDR}$, differing from the transfer function in essence. The proposed GMNet is inspired by the latest HDR data format [1] [2] [3], and the novelty of exploring this new format has been uniformly recognized by the other three reviewers.
>
> [1] ISO. "Gain map metadata for image conversion." https://www.iso.org/standard/86775.html, 2024.
>
> [2] Adobe. "Gainmap specification." https://helpx.adobe.com/camera-raw/using/gain-map.html, 2024.
>
> [3] Google. "Ultra-hdr image format." https://developer.android.com/media/platform/hdr-image-format, 2024.
>
>
>
> **2. Additional comparison methods, e.g., DCDR-UNet**
>
> We keep up with the latest research, but there have been few open-source methods in recent years. For example, the reviewer mentioned DCDR-UNet [4] (NOT open-sourced yet), in which the latest comparison method is also KUNet that we have already compared in the paper. Besides, we have tried to positively contact the authors for code during the research and rebuttal periods, but we did not get responses.
>
> To address reviewers' concerns, we unofficially reproduce DCDR-UNet by ourselves and add two more baselines, EPCE-HDR [5] and ITM-LUT [6]. The experiment results on the synthetic dataset are shown below, demonstrating that our method still achieves superior performance over these latest works. Finally, we guarantee that our work will be open-sourced to promote research along this line.
>
> |   ㅤMethodㅤ    |   PSNR-L↑   |  SSIM-L↑   |  PSNR-PQ↑   |  SSIM-PQ↑  | $ΔE_{ITP}$↓ | HDR-VDP3↑  |
> | :---------: | :---------: | :--------: | :---------: | :--------: | :---------: | :--------: |
> |    KUNet    |   40.4879   |   0.9952   |   34.5201   |   0.9619   |   7.1538    |   9.7851   |
> |   HDRUNet   |   41.2579   |   0.9969   |   40.9315   |   0.9946   |   3.5799    |   9.9080   |
> |    HDCFM    |   41.8563   |   0.9967   |   44.9017   |   0.9992   |   2.4773    |   9.9115   |
> |    FMNet    |   40.7699   |   0.9970   |   44.4798   |   0.9996   |   2.3032    |   9.8991   |
> | *DCDR-UNet* |   41.8085   |   0.9971   |   42.0183   |   0.9962   |   3.0501    |   9.9215   |
> | *EPCE-HDR*  |   41.5919   |   0.9956   |   31.4700   |   0.9430   |   8.5830    |   9.6694   |
> |  *ITM-LUT*  |   39.5735   |   0.9950   |   43.7152   |   0.9987   |   2.2865    |   9.7268   |
> |    Ours     | **43.5510** | **0.9977** | **47.6256** | **0.9998** | **1.6262**  | **9.9477** |
>
> [4] Kim, Joonsoo, et al. "DCDR-UNet: Deformable Convolution Based Detail Restoration via U-shape Network for Single Image HDR Reconstruction." *Proceedings of the IEEE/CVF Conference on Computer Vision and Pattern Recognition*. 2024.
>
> [5] Tang, Jiaqi, et al. "High Dynamic Range Image Reconstruction via Deep Explicit Polynomial Curve Estimation." *ECAI 2023*. IOS Press, 2023. 2330-2337.
>
> [6] Guo, Cheng, et al. "Redistributing the Precision and Content in 3D-LUT-based Inverse Tone-mapping for HDR/WCG Display." *Proceedings of the 20th ACM SIGGRAPH European Conference on Visual Media Production*. 2023.

---

> > ### Comment · Reviewer_6tzz · 2024-11-26
> >
> > The reviewer values this method, however although the formulations are different between transfer function and mapping, the idea behind it is still similar in which every pixel in LDR domain has its own corresponding value in HDR domain. that is why the reviewer doesn't feel convinced about the novelty of this paper.
> >
> > Furthermore, even though the code for DCDR-UNet is not publicly available, it is worth mentioning it in the literature reviews.
> >
> > Finally, the evaluation should be done on both synthetic and real data, to avoid the cherry-picking results.

---

> > > ### Author Response · Authors · 2024-11-26
> > >
> > > Thank you for providing further feedback. We are encouraged that “the reviewer values this method”, and we would like to address the remaining concerns as below:
> > >
> > > **1. On the novelty of our paper**
> > >
> > > We respectfully argue the justification here.
> > >
> > > The reviewer agrees that “the formulations are different between transfer function and mapping”. As **the first attempt** to introduce GM to the ITM task, we firmly believe the proposed method itself carves out a new direction for this task and could inspire future works along this line. This is also a consensus among other reviewers.
> > >
> > > Moreover, it is NOT reasonable to dismiss the novelty of our work since “the idea behind it is still similar (to mapping)“, since this vague argument can be applied almost everywhere.
> > >
> > > Even for the mapping direction, a number of methods based on mapping have been proposed [1] [2] [3] [4] [5] [6], and their novelty has been widely recognized. It is thus clear that **whether a method is based on mapping should NOT be regarded as a measure of novelty.**
> > >
> > > **2. On the discussion of DCDR-UNet**
> > >
> > > We cited DCDR-UNet in the original manuscript. As suggested, we provided the comparison results in the last revised version. For additional literature review, we have added a detailed discussion for DCDR-UNet in the Related Work Section in the latest version.
> > >
> > > **3. On the evaluation of real-world dataset**
> > >
> > > We have already provided the real-world results in the last revised version (see Table 3 in the main paper). For the convenience of review, we again list the experimental results on the real-world dataset below. As can be seen, the advantages of the proposed method remain on most metrics (the only exception is HDR-VDP3, where ours is very near to DCDR-UNet).
> > >
> > > |   ㅤMethodㅤ    |   PSNR-L↑   |  SSIM-L↑   |  PSNR-PQ↑   |  SSIM-PQ↑  | $ΔE_{ITP}$↓ | HDR-VDP3↑  |
> > > | :---------: | :---------: | :--------: | :---------: | :--------: | :---------: | :--------: |
> > > |    KUNet    |   33.3256   |   0.9907   |   33.7895   |   0.9690   |   9.0578    |   9.6893   |
> > > |   HDRUNet   |   33.1686   |   0.9911   |   37.3966   |   0.9969   |   5.7358    |   9.8455   |
> > > |    HDCFM    |   32.7963   |   0.9879   |   38.1894   |   0.9956   |   6.2957    |   9.8035   |
> > > |    FMNet    |   32.6247   |   0.9921   |   39.7632   |   0.9988   |   4.5544    |   9.8110   |
> > > | *DCDR-UNet* |   33.5421   |   0.9924   |   38.3252   |   0.9970   |   4.8645    | **9.8781** |
> > > | *EPCE-HDR*  |   32.7616   |   0.9914   |   31.1207   |   0.9466   |   10.4322   |   9.7218   |
> > > |  *ITM-LUT*  |   31.6677   |   0.9842   |   37.9021   |   0.9958   |   5.4299    |   9.6831   |
> > > |    Ours     | **33.9490** | **0.9928** | **40.2088** | **0.9993** | **4.0260**  |   9.8757   |
> > >
> > > **In summary, we believe the major concerns are addressed in the latest manuscript by integrating valuable suggestions from all expert reviewers, and we hope the strength of our work can be better recognized now.**
> > >
> > > Thanks again for your valuable time and we look forward to your update on whether the remaining concerns have been addressed.
> > >
> > >  ㅤ
> > >
> > > [1] Kong, Lingtong, et al. "SAFNet: Selective Alignment Fusion Network for Efficient HDR Imaging." *European Conference on Computer Vision*. Springer, Cham, 2024.
> > >
> > > [2] Tel, Steven, et al. "Alignment-free hdr deghosting with semantics consistent transformer." *Proceedings of the IEEE/CVF International Conference on Computer Vision.* 2023.
> > >
> > > [3] Chung, Haesoo, and Nam Ik Cho. "Lan-hdr: Luminance-based alignment network for high dynamic range video reconstruction." *Proceedings of the IEEE/CVF International Conference on Computer Vision*. 2023.
> > >
> > > [4] Xu, Gangwei, et al. "HDRFlow: Real-Time HDR Video Reconstruction with Large Motions." *Proceedings of the IEEE/CVF Conference on Computer Vision and Pattern Recognition*. 2024.
> > >
> > > [5] Shu, Yong, et al. "Towards Real-World HDR Video Reconstruction: A Large-Scale Benchmark Dataset and A Two-Stage Alignment Network." *Proceedings of the IEEE/CVF Conference on Computer Vision and Pattern Recognition*. 2024.
> > >
> > > [6] Kim, Joonsoo, et al. "DCDR-UNet: Deformable Convolution Based Detail Restoration via U-shape Network for Single Image HDR Reconstruction." *Proceedings of the IEEE/CVF Conference on Computer Vision and Pattern Recognition*. 2024.

---

> > > ### Author Response · Authors · 2024-11-28
> > >
> > > Dear Reviewer 6tzz,
> > >
> > > Thanks again for your valuable time. We are honored to see that with the advice from all expert reviewers, our paper has been further improved and better recognized by now.
> > >
> > > As today is the last day the authors can upload a revised PDF, please feel free to let us know if you have any further questions. We are glad to provide further details on any aspects of our responses that may require additional clarification or elaboration.
> > >
> > > We look forward to your positive feedback if your major concerns have been adequately addressed.
> > >
> > > Sincerely,
> > >
> > > The authors

---

> > > > ### Comment · Reviewer_6tzz · 2024-11-29
> > > >
> > > > The author has addressed all of my concerns, I'm happy to update my score.

---

> > > > > ### Author Response · Authors · 2024-11-30
> > > > >
> > > > > Thank you for raising the score.
> > > > >
> > > > > We appreciate your thoughtful feedback and the time you took to review our manuscript, and we are grateful for your recognition of the revisions and improvements made to address your concerns. Thank you again for your efforts in helping us enhance the quality of our work!

---

> ### Author Response · Authors · 2024-11-21
>
> **3. Concerns on the metrics of PSNR-NGM and PSNR-mu**
>
> First, we need to clarify that PSNR-NGM is NOT a metric for evaluating HDR images. It is only used in the ablation experiments on our network, evaluating the quality of the normalized GM.
>
> Second, we follow the previous work [7] using PSNR-L in the linear domain, but employ PSNR-PQ to evaluate visual similarity instead of PSNR-mu, which plays a similar role in compressing HDR images. The main reason is that our work is closer to ITM tasks, so applying PQ-metric is more suitable for practical applications.
>
> Third, to further address the reviewer's concerns, we further provide the experimental results of PSNR-mu, PSNR-PQ, and PSNR-L on real-world datasets in the table below, and our method still achieves superior performance.
>
> |   ㅤMethodㅤ    |  PSNR-mu↑   |  PSNR-PQ↑   |   PSNR-L↑   |
> | :---------: | :---------: | :---------: | :---------: |
> |    KUNet    |   33.2771   |   33.7895   |   33.3256   |
> |   HDRUNet   |   37.5750   |   37.3966   |   33.1686   |
> |    HDCFM    |   38.7144   |   38.1894   |   32.7963   |
> |    FMNet    |   41.0164   |   39.7632   |   32.6247   |
> | *DCDR-UNet* |   39.1466   |   38.3252   |   33.5421   |
> | *EPCE-HDR*  |   30.3388   |   31.1207   |   32.7616   |
> |  *ITM-LUT*  |   39.2880   |   37.9021   |   31.6677   |
> |    Ours     | **41.6012** | **40.2088** | **33.9490** |
>
> [7] Chen, Xiangyu, et al. "Hdrunet: Single image hdr reconstruction with denoising and dequantization." *Proceedings of the IEEE/CVF Conference on Computer Vision and Pattern Recognition*. 2021.
>
>
>
> **4. Explanation of the normalization of $Q_{max}$**
>
> The $Q_{max}$ in the real-world dataset is in the range of [0,5], and the $I_{GM}$ is also in the range of [0,5]. The $I_{GM}$ is divided by the **global** maximum **5** for normalization, differing from the $I_{NGM}$ normalized by the **respective** maximum of each data. Therefore, $I_{GM}$ will be normalized to [0,1], making a uniform and smooth convergence.
>
>
>
> **5. Concerns on the upsampling of $I_{GM}$**
>
> There might be some misunderstanding here. The resolution of GM in the real world is reduced in most cases, as the downsampling is recognized by the standards [1] [2] [3] to save the bandwidth. Therefore, our method aims to learn the **Groud-Turth** GM, not the **Interpolated** GM.
>
> Moreover, we perform ablation experiments on the resolution of GM in Table 5 in our paper, demonstrating that GMNet can achieve superior performance no matter with or without interpolation.

---

### Official Review · Reviewer_nA6K · 2024-11-03

**Soundness:** 4
**Presentation:** 3
**Contribution:** 4
**Rating:** 6
**Confidence:** 4

**Summary:**

With recent advancements in display technology and the widespread adoption of HDR displays, a new dual-layer image format has emerged that is compatible with existing SDR displays. In response to this, the authors propose a new type of inverse tone mapping algorithm that infers a gain map applicable to adaptive HDR displays. Additionally, the authors construct an additional dataset for training and evaluating this type of network. Through experiments, the authors demonstrate that their method restores HDR images more effectively and reliably compared to other approaches.

**Strengths:**

* The authors' proposed inverse tone mapping algorithm, which takes real display environments into account, is both highly practical and innovative.
* The authors have effectively organized the existing methods in this field through Table 1. I believe this will be very helpful for future researchers studying this area.
* They have provided a formulation for the proposed method, which makes it easier to understand the approach proposed by the authors.

**Weaknesses:**

* The authors need to conduct additional survey on SI-HDR. According to [A], SI-ITM is broadly divided into two branches of research: 1) methods that directly reconstruct HDR images and 2) methods that reconstruct bracketed exposures. In my opinion, there is a need to provide a detailed description of the differences between Learning from LDR stacks methods and generating a gain map.

* Furthermore, the authors lack comparisons not only in terms of performance for learning from LDR stacks but also in terms of time efficiency with existing methods.
  * Like [B], it is possible to create images with relative EV +1 / -1 and expand the dynamic range. A discussion should be added on whether GM-ITM is more efficient than this case or not.

* As the authors propose a new type of network, they need to provide a detailed structure of this network.

* There is a lack of validation on datasets commonly used in inverse tone mapping.
   - HDR-SYNTH + HDR-REAL: Yu-Lun Liu et al. Single-image hdr reconstruction by learning to reverse the camera pipeline. CVPR, 2020
   - HDREye: Hiromi Nemoto et al. Visual attention in ldr and hdr images. 9th International Workshop on Video Processing and Quality Metrics for Consumer Electronics (VPQM), 2015.
  - VDS: Siyeong Lee et al. Deep chain HDRI: Reconstructing a high dynamic range image from a single low dynamic range image. IEEE Access, 2018

[A] Lin Wang and Kuk-Jin Yoon, Deep Learning for HDR Imaging: State-of-the-Art and Future Trends, TPAMI, https://arxiv.org/abs/2110.10394

[B] Ning Zhang et al. Revisiting the Stack-Based Inverse Tone Mapping, CVPR, 2023

**Questions:**

* Could the authors explain why the GM can be transmitted at a reduced resolution compared to the original SDR?
* Could you share experimental results for directly learning Q_max adjusted for the number of pixels?
* Would it be possible to release the code and data to ensure reproducibility?

* If I have misunderstood any part or if my concerns are adequately addressed during the review process, I would be more than willing to increase my score.

---

> ### Author Response · Authors · 2024-11-21
>
> We appreciate the reviewer's valuable comments and suggestions, and we hope our responses could address the concerns.
>
> **1. Additional investigation on stack-based methods**
>
> The stack-based HDR reconstruction task mentioned by the reviewer is an important research branch of SI-HDR. We investigate the works of [1] [2] [3] [4] [5] [6], and summarize the main differences from our method as follows. The GM-ITM methods are inspired by a novel HDR image format, learning auxiliary data GM that can upgrade SDR to HDR display. By contrast, the stack-based methods simulate the HDRI technology, generating the multi-exposure stack for HDR reconstruction. Besides, compared to the stack-based methods, the GM-ITM methods obtain the final HDR result by multiplication of a single SDR image and the corresponding GM, avoiding the artifacts that may occur during the multi-image fusion process. We will include these discussions in the revised paper.
>
>
> [1] Wang, Lin, and Kuk-Jin Yoon. "Deep learning for hdr imaging: State-of-the-art and future trends." *IEEE transactions on pattern analysis and machine intelligence* 44.12 (2021): 8874-8895.
>
> [2] Endo, Yuki, Yoshihiro Kanamori, and Jun Mitani. "Deep reverse tone mapping." *ACM Trans. Graph* 36.6 (2017): 1-10.
>
> [3] Lee, Siyeong, Gwon Hwan An, and Suk-Ju Kang. "Deep chain hdri: Reconstructing a high dynamic range image from a single low dynamic range image." *IEEE Access* 6 (2018): 49913-49924.
>
> [4] Lee, Siyeong, Gwon Hwan An, and Suk-Ju Kang. "Deep recursive hdri: Inverse tone mapping using generative adversarial networks." *proceedings of the European Conference on Computer Vision (ECCV)*. 2018.
>
> [5] Kim, Junghee, Siyeong Lee, and Suk-Ju Kang. "End-to-end differentiable learning to HDR image synthesis for multi-exposure images." *Proceedings of the AAAI Conference on Artificial Intelligence*. Vol. 35. No. 2. 2021.
>
> [6] Zhang, Ning, et al. "Revisiting the stack-based inverse tone mapping." *Proceedings of the IEEE/CVF Conference on Computer Vision and Pattern Recognition*. 2023.
>
>
>
> **2. Comparison experiments on efficiency**
>
> We actively contacted the authors of [6] for the code but did not get responses. Therefore, we compare the efficiency of our approach with other available methods [7] [8] [9], and the experimental results are shown below. The runtime is evaluated in NVIDIA A100 as the average of 100 trials on the real-world dataset in the resolution of $4096\times3072$.
>
> |   ㅤMethodㅤ    | Params↓ |  ㅤRuntime↓ㅤ  |   ㅤFLOPs↓ㅤ   |   PSNR-L↑   |  PSNR-PQ↑   | $ΔE_{ITP}$↓ |
> | :---------: | :-----: | :--------: | :--------: | :---------: | :---------: | :---------: |
> |    KUNet    |  1.08M  |  621.70ms  |  7534.50G  |   33.3256   |   33.7895   |   9.0578    |
> |   HDRUNet   |  1.58M  |  613.89ms  |  4161.75G  |   33.1686   |   37.3966   |   5.7358    |
> |    HDCFM    |  0.10M  |  497.05ms  |  127.08G   |   32.7963   |   38.1894   |   6.2957    |
> |    FMNet    |  1.24M  |  304.13ms  |  4147.13G  |   32.6247   |   39.7632   |   4.5544    |
> | *DCDR-UNet* |  1.26M  | 1085.91ms  |  4502.77G  |   33.5421   |   38.3252   |   4.8645    |
> | *EPCE-HDR*  | 31.02M  | 23992.84ms | 326412.14G |   32.7616   |   31.1207   |   10.4322   |
> |  *ITM-LUT*  |  0.57M  |  19.81ms   |   59.16G   |   31.6677   |   37.9021   |   5.4299    |
> |    Ours     |  1.83M  |  75.26ms   |  1112.18G  | **33.9490** | **40.2088** | **4.0260**  |
>
> As can be seen, benefiting from the simple form of the target GM, our method achieves the second fastest runtime. While the look-up-table-based solution ITM-LUT is faster, its performance on image quality metrics is much lower.
>
> [7] Kim, Joonsoo, et al. "DCDR-UNet: Deformable Convolution Based Detail Restoration via U-shape Network for Single Image HDR Reconstruction." *Proceedings of the IEEE/CVF Conference on Computer Vision and Pattern Recognition*. 2024.
>
> [8] Tang, Jiaqi, et al. "High Dynamic Range Image Reconstruction via Deep Explicit Polynomial Curve Estimation." *ECAI 2023*. IOS Press, 2023. 2330-2337.
>
> [9] Guo, Cheng, et al. "Redistributing the Precision and Content in 3D-LUT-based Inverse Tone-mapping for HDR/WCG Display." *Proceedings of the 20th ACM SIGGRAPH European Conference on Visual Media Production*. 2023.
>
> **3. Detailed structure of the proposed network**
>
> We will provide more details of our network structure in the revised version of our paper. Specifically, in the local head, we use three convolutional layers and ReLU activation functions to extract initial features, where all convolutional layers are in size 3, stride 1, and padding 1, except for the stride of the first convolutional layer is set to 2 to downsample the input. The local tail can be represented as $(Conv \circ ReLU)^2  \circ PS \circ Conv$, where $(\cdot)^n$ means cascade of $n$ modules and all convolutional layers are in size 3, stride 1, and padding 1. The rest implementation details of the other modules can be found in our code that will be open-sourced.

---

> ### Author Response · Authors · 2024-11-21
>
> **4. Validation experiments on other datasets**
>
> Thank you for providing information of additional datasets. We would like to first explain that these datasets are customized for the SI-HDR task, which is different from the intent of the ITM task. The ITM task focuses more on dynamic range restoration and color gamut conversion instead of the illusion of missing details and reconstruction of unlimited irradiance. Nevertheless, as suggested, we conduct additional experiments and the results for VDS [3] and HDREye [10] are shown below.
>
> | Method |   PSNR-L↑   |  SSIM-L↑   |  PSNR-PQ↑   |  SSIM-PQ↑  |  $ΔE_{ITP}$↓   |
> | :----: | :---------: | :--------: | :---------: | :--------: | :---------: |
> | KUNet  |   29.0394   |   0.8435   |   19.9152   |   0.8020   |   44.6197   |
> | FMNet  | **29.0543** |   0.8447   |   20.1409   |   0.8282   |   43.7337   |
> |  Ours  |   28.8087   | **0.8468** | **21.1474** | **0.8356** | **40.8171** |
>
>
>
> | Method |   PSNR-L↑   |  SSIM-L↑   |  PSNR-PQ↑   |  SSIM-PQ↑  |   $ΔE_{ITP}$↓   |
> | :----: | :---------: | :--------: | :---------: | :--------: | :---------: |
> | KUNet  |   23.1648   |   0.6641   |   15.5961   |   0.5398   |   64.0899   |
> | FMNet  |   23.2416   |   0.6622   |   15.9804   |   0.5929   |   62.1801   |
> |  Ours  | **23.2446** | **0.6758** | **16.6180** | **0.6023** | **56.7420** |
>
> Note that, while the domain gap between the tasks makes the quantitative metrics relatively low for all methods, the qualitative comparisons of diverse data from different sources [3] [10] [11] are meaningful and help to increase the credibility of our experiments. The results of the qualitative experiments are shown below.
>
> [Figure A] https://picx.zhimg.com/80/v2-b62509ce89fd10e0a848910fa2465d4e.png
>
> [Figure B] https://picx.zhimg.com/80/v2-e0543ac6ca186abb25a5b05ac38e92a3.png
>
> Experimental results show that our method recovers local and global contrast well and achieves smooth and realistic visual results, verifying its generalizability over a wider range of data. We will add these additional results in the revised paper or supplementary document.
>
>
>
> [10] Nemoto, Hiromi, et al. "Visual attention in LDR and HDR images." *9th International Workshop on Video Processing and Quality Metrics for Consumer Electronics (VPQM)*. 2015.
>
> [11] Liu, Yu-Lun, et al. "Single-image HDR reconstruction by learning to reverse the camera pipeline." *Proceedings of the IEEE/CVF conference on computer vision and pattern recognition*. 2020.
>
>
>
>
>
> **5. Why the GM can be transmitted at a reduced resolution?**
>
> The downsampling of the GM is recognized by standards [12] [13] [14] to save the bandwidth. The resolution-sensitive information such as edges, textures, etc., is stored in the full-resolution SDR image, and the dynamic range information recorded by GM is coupled with the SDR image. Therefore, the downsampling has a small impact on the detail of the final HDR, which is acceptable for bandwidth saving.
>
> [12] ISO. "Gain map metadata for image conversion." https://www.iso.org/standard/86775.html, 2024.
>
> [13] Adobe. "Gainmap specification." https://helpx.adobe.com/camera-raw/using/gain-map.html, 2024.
>
> [14] Google. "Ultra-hdr image format." https://developer.android.com/media/platform/hdr-image-format, 2024.
>
>
>
> **6. Experimental results for directly learning $Q_{max}$**
>
> The ablation results on the synthesis dataset are shown in the table below, where "-", "O", and "√" in the QM column represent not learning, directly learning, and indirectly learning, respectively. Compared to not learning $Q_{max}$, learning $Q_{max}$ directly gets better linear performance, but the imbalanced supervision between tensor and scalar causes a decline in perception metrics, and all the results have a significant decay compared to indirectly learning $Q_{max}$.
>
> | Backbone |  QM   |  CW   |  SA   |   PSNR-L↑   |  PSNR-PQ↑   | HDR-VDP3↑  |
> | :------: | :---: | :---: | :---: | :---------: | :---------: | :--------: |
> |    √     |   -   |   -   |   -   |   41.1670   |   45.2765   |   9.9210   |
> |  **√**   | **O** | **-** | **-** | **41.3755** | **44.9717** | **9.9128** |
> |    √     |   √   |   -   |   -   |   42.3479   |   46.2067   |   9.9431   |
> |    √     |   √   |   √   |   -   |   43.0480   |   47.1238   |   9.9454   |
> |    √     |   √   |   -   |   √   |   42.9085   |   47.0011   |   9.9442   |
> |    √     |   √   |   √   |   √   |   43.5510   |   47.6256   |   9.9477   |
>
>
>
> **7. Open source for code and datasets**
>
> We commit to making our codes and datasets publicly available upon acceptance of the paper, enabling the researchers to validate our work and apply it to broader scenarios. We appreciate your emphasis on transparency and look forward to contributing to the community in this way.

---

> > ### Comment · Reviewer_nA6K · 2024-11-22
> > **Question about "Validation experiments on other datasets"**
> >
> > Could you tell me which exposure values were selected as input images from the multi-exposure stacks in VDS/HDREye? Additionally, could you provide the average HDR-VDP-3 (or HDR-VDP-2) metrics for the generated HDR images?

---

> ### Author Response · Authors · 2024-11-22
>
> We select LDR images with the exposure of 0 [EV] as the inputs for both VDS and HDREye datasets. The experiments with additional HDR-VDP3 metric on VDS and HDREye datasets are shown below.
>
> | Method |   PSNR-L↑   |  SSIM-L↑   |  PSNR-PQ↑   |  SSIM-PQ↑  | $ΔE_{ITP}$↓ | HDR-VDP3↑  |
> | :----: | :---------: | :--------: | :---------: | :--------: | :---------: | :--------: |
> | KUNet  |   29.0394   |   0.8435   |   19.9152   |   0.8020   |   44.6197   |   6.7761   |
> | FMNet  | **29.0543** |   0.8447   |   20.1409   |   0.8282   |   43.7337   | **6.7819** |
> |  Ours  |   28.8087   | **0.8468** | **21.1474** | **0.8356** | **40.8171** |   6.5656   |
>
>
>
> | Method |   PSNR-L↑   |  SSIM-L↑   |  PSNR-PQ↑   |  SSIM-PQ↑  | $ΔE_{ITP}$↓ | HDR-VDP3↑  |
> | :----: | :---------: | :--------: | :---------: | :--------: | :---------: | :--------: |
> | KUNet  |   23.1648   |   0.6641   |   15.5961   |   0.5398   |   64.0899   |   5.5913   |
> | FMNet  |   23.2416   |   0.6622   |   15.9804   |   0.5929   |   62.1801   | **5.7408** |
> |  Ours  | **23.2446** | **0.6758** | **16.6180** | **0.6023** | **56.7420** |   5.7035   |
>
> As mentioned, these datasets are customized for the Sl-HDR task with a different intent from the ITM task. **The uniformly low quantitative metrics for all methods clearly demonstrate the domain gap between the two tasks, making the comparison results less informative.**

---

> > ### Comment · Reviewer_nA6K · 2024-11-26
> >
> > My concerns have been fully addressed by the experiments added to the review, so I am upgrading the score. Additionally, I would like to see some quantitative metrics added to explain why the performance degradation is attributed to the domain gap.

---

> > > ### Author Response · Authors · 2024-11-26
> > >
> > > Thank you for raising the score.
> > >
> > > We are encouraged that our response has effectively addressed your concerns. We sincerely appreciate your thorough review and the valuable time you have taken to help us improve our work.
> > >
> > > Regarding your request for quantitative metrics to explain the performance degradation attributed to the domain gap, we appreciate the suggestion and agree that this would provide valuable insights. We are currently working on it and need a bit more time to complete the experiments. We aim to provide these additional results as soon as possible. Thank you again for your thoughtful feedback!

---

> > > ### Author Response · Authors · 2024-11-30
> > >
> > > Dear reviewer nA6K,
> > >
> > > Thank you for your generous support first. Regarding your request for quantitative metrics to explain the performance degradation attributed to the domain gap, we make the following clarifications.
> > >
> > > As stated in [A] [B], the HDR images in SI-HDR datasets are suggested to be reproduced with physically correct values using measured data. However, most HDR images in **SI-HDR datasets** do not have this data and store relative irradiance values, while **GM-ITM datasets** represent absolute display values. For a fair comparison, we choose the following quantitative metrics that do not rely on absolute values to measure the gap between datasets.
> > >
> > > |      ㅤㅤㅤDatasetㅤㅤㅤ       | $CV_{peak}$ | Saturation | Kurtosis | Skewness |
> > > | :----------------: | :---------: | :--------: | :------: | :------: |
> > > |    SI-HDR (VDS)    |   1.4080    |   0.3710   | 81.8410  |  4.4959  |
> > > |  SI-HDR (HDREye)   |   1.6477    |   0.3275   | 33.0952  |  3.9520  |
> > > | GM-ITM (Synthetic) |   0.4726    |   0.6030   | 11.9692  |  2.1164  |
> > > | GM-ITM (Real-wold) |   0.2880    |   0.4756   |  7.6836  |  2.0965  |
> > >
> > > (1) Peak Variation. The **Coefficient of Variation (CV) of peak value** varies significantly between the SI-HDR and GM-ITM datasets. The SI-HDR data records real-world irradiance and the peaks fluctuate widely due to the diverse scenes, while the GM-ITM data is display-referred, in which the peaks are smoother for a better visual experience.
> > >
> > > (2) Colorfulness. The SI-HDR data stores irradiance that has not been post-processed, so the **Saturation** is lower than GM-ITM data intended for display.
> > >
> > > (3) Distribution. To better utilize the display capabilities of the device, GM-ITM data exhibits more balanced. In contrast, the SI-HDR data stores real-world irradiance that varies widely, thus the **Kurtosis** and **Skewness** are more extreme, making a significant gap in the distribution with the GM-ITM data.
> > >
> > > We hope your remaining concerns can be adequately addressed now. If possible, we look forward to more positive feedback. Thanks once again!
> > >
> > > ㅤ
> > >
> > > [A] Nemoto, Hiromi, et al. "Visual attention in LDR and HDR images." *9th International Workshop on Video Processing and Quality Metrics for Consumer Electronics (VPQM)*. 2015.
> > >
> > > [B] Akyüz, Ahmet Oǧuz, et al. "Do HDR displays support LDR content? A psychophysical evaluation." *ACM Transactions on Graphics (TOG)*. 2007.

---

### Official Review · Reviewer_cLvc · 2024-11-03

**Soundness:** 3
**Presentation:** 3
**Contribution:** 3
**Rating:** 8
**Confidence:** 3

**Summary:**

The paper introduces a new task, Gain Map-based Inverse Tone Mapping (GM-ITM), to address the problem of converting standard dynamic range (SDR) images into high dynamic range (HDR) images using a Gain Map (GM). This work aims to improve HDR up-conversion by focusing on GM estimation instead of directly predicting HDR, leveraging a dual-branch network (GMNet) that includes Local Contrast Restoration (LCR) and Global Luminance Estimation (GLE) branches. To support the research, the authors also propose synthetic and real-world datasets to evaluate the method. The experiments demonstrate GMNet’s quantitative and qualitative superiority over existing methods for HDR-related tasks.

**Strengths:**

Novelty: The paper addresses an emerging area in HDR up-conversion by proposing the GM-ITM task, which is relatively unexplored and offers an innovative approach to inverse tone mapping.

Method: The proposed dual-branch GMNet architecture is well-designed, with LCR and GLE branches targeting local and global image features respectively, which allows for improved GM estimation.

Evaluation: The authors perform extensive quantitative and qualitative evaluations on synthetic and real-world datasets, benchmarking GMNet against well-established HDR methods and showing clear improvements.

Dataset: The creation of both synthetic and real-world datasets for GM-ITM facilitates further research and addresses a gap in data availability, providing a valuable resource for the field.

Ablation Studies: The paper includes thorough ablation studies on key components, such as spatial-aware modulation and GM resolution, to substantiate the effectiveness of GMNet’s design choices.

**Weaknesses:**

Limited Comparison Scope: The paper primarily compares GMNet with existing HDR and SDR-to-HDRTV up-conversion methods, but a comparison with more diverse or advanced inverse tone mapping techniques could strengthen the results.

Complexity and Computation: The dual-branch network adds computational overhead, especially when processing high-resolution images. Discussion on efficiency or real-time applicability is limited, which could impact practical usage.

Dependency on New Data Format: The proposed method relies heavily on the novel GM format, which is not yet widely adopted. This dependency could limit the method's applicability outside specialized devices or formats.

Visual Comparisons in Real-World Scenarios: Although qualitative results demonstrate GMNet’s advantages, additional challenging real-world scenarios could better highlight its robustness. For instance, extreme lighting conditions or complex textures may expose limitations.

**Questions:**

A more detailed discussion about the computation complexity would help understand the applicability of the proposed method.

---

> ### Author Response · Authors · 2024-11-21
>
> We appreciate the reviewer's valuable comments and suggestions, and we hope our responses could address the concerns.
>
> **1. Comparison with more related methods**
>
> To further increase the comprehensiveness of the experiments, we append three more baselines for comparisons, namely DCDR-UNet [1], EPCE-HDR [2], and ITM-LUT [3]. The experiment results on the synthetic dataset are shown below, demonstrating that our method still achieves superior performance over these latest works.
>
> |   ㅤMethodㅤ    |   PSNR-L↑   |  SSIM-L↑   |  PSNR-PQ↑   |  SSIM-PQ↑  | $ΔE_{ITP}$↓ | HDR-VDP3↑  |
> | :---------: | :---------: | :--------: | :---------: | :--------: | :---------: | :--------: |
> |    KUNet    |   40.4879   |   0.9952   |   34.5201   |   0.9619   |   7.1538    |   9.7851   |
> |   HDRUNet   |   41.2579   |   0.9969   |   40.9315   |   0.9946   |   3.5799    |   9.9080   |
> |    HDCFM    |   41.8563   |   0.9967   |   44.9017   |   0.9992   |   2.4773    |   9.9115   |
> |    FMNet    |   40.7699   |   0.9970   |   44.4798   |   0.9996   |   2.3032    |   9.8991   |
> | *DCDR-UNet* |   41.8085   |   0.9971   |   42.0183   |   0.9962   |   3.0501    |   9.9215   |
> | *EPCE-HDR*  |   41.5919   |   0.9956   |   31.4700   |   0.9430   |   8.5830    |   9.6694   |
> |  *ITM-LUT*  |   39.5735   |   0.9950   |   43.7152   |   0.9987   |   2.2865    |   9.7268   |
> |    Ours     | **43.5510** | **0.9977** | **47.6256** | **0.9998** | **1.6262**  | **9.9477** |
>
> [1] Kim, Joonsoo, et al. "DCDR-UNet: Deformable Convolution Based Detail Restoration via U-shape Network for Single Image HDR Reconstruction." *Proceedings of the IEEE/CVF Conference on Computer Vision and Pattern Recognition*. 2024.
>
> [2] Tang, Jiaqi, et al. "High Dynamic Range Image Reconstruction via Deep Explicit Polynomial Curve Estimation." *ECAI 2023*. IOS Press, 2023. 2330-2337.
>
> [3] Guo, Cheng, et al. "Redistributing the Precision and Content in 3D-LUT-based Inverse Tone-mapping for HDR/WCG Display." *Proceedings of the 20th ACM SIGGRAPH European Conference on Visual Media Production*. 2023.
>
>
>
> **2. Complexity and computation**
>
> There is no additional burden on our dual-branch network for high-resolution input, as the input of the GLE branch $I_{SDR}^{LR}$ is at a fixed $256\times256$ resolution. To evaluate the computational efficiency and real-time processing capability of our method, we conduct comparison experiments and add three more baselines [1] [2] [3]. The runtime is evaluated in NVIDIA A100 as the average of 100 trials on the real-world dataset in the resolution of $4096\times3072$.
>
> |   ㅤMethodㅤ   | Params↓ |  ㅤRuntime↓ㅤ  |   ㅤFLOPs↓ㅤ   |   PSNR-L↑   |  PSNR-PQ↑   | $ΔE_{ITP}$↓ |
> | :---------: | :-----: | :--------: | :--------: | :---------: | :---------: | :---------: |
> |    KUNet    |  1.08M  |  621.70ms  |  7534.50G  |   33.3256   |   33.7895   |   9.0578    |
> |   HDRUNet   |  1.58M  |  613.89ms  |  4161.75G  |   33.1686   |   37.3966   |   5.7358    |
> |    HDCFM    |  0.10M  |  497.05ms  |  127.08G   |   32.7963   |   38.1894   |   6.2957    |
> |    FMNet    |  1.24M  |  304.13ms  |  4147.13G  |   32.6247   |   39.7632   |   4.5544    |
> | *DCDR-UNet* |  1.26M  | 1085.91ms  |  4502.77G  |   33.5421   |   38.3252   |   4.8645    |
> | *EPCE-HDR*  | 31.02M  | 23992.84ms | 326412.14G |   32.7616   |   31.1207   |   10.4322   |
> |  *ITM-LUT*  |  0.57M  |  19.81ms   |   59.16G   |   31.6677   |   37.9021   |   5.4299    |
> |    Ours     |  1.83M  |  75.26ms   |  1112.18G  | **33.9490** | **40.2088** | **4.0260**  |
>
> As can be seen, benefiting from the simple form of the target GM, our method achieves the second fastest runtime. While the look-up-table-based solution ITM-LUT is faster, its performance on image quality metrics is much lower.

---

> ### Author Response · Authors · 2024-11-21
>
> **3. Relevance to the new data format**
>
> It needs to be clarified that our method does not rely on the new GM format. The proposed GMNet can estimate GM from the input SDR image. Then the SDR-GM pair can be directly encapsulated into the new GM format, but also can be calculated to linear HDR through the pipeline in Section 3. After that, we can convert the linear HDR into other mainstream HDR formats, such as PQ-encoded HDR image in HDR10 standard, not limited to specialized devices or formats.
>
>
>
> **4. Visual comparisons in challenging real-world scenarios**
>
> Figure 9 and Figure 10 in the Appendix demonstrate the superiority of our method in reconstructing edges and high-contrast night scenes. To enable more in-depth analysis in challenging scenarios, we perform qualitative experiments as follows.
>
> [Figure A] https://picx.zhimg.com/80/v2-f7a1fc6a62f5bf91169047c041d40f49.png
>
> [Figure B] https://picx.zhimg.com/80/v2-9a63b9bf1e886282ae886f41fccfad9e.png
>
> As shown in Figure A, our method achieves superior performance in the sun region at extreme brightness. The HDCFM also achieves low errors, but with grid effect due to operator properties. Figure B shows the estimation results on the leaves with complex textures and eaves with sharp edges, and our method demonstrate superior performance on complex textures in the real-world scenario, validating the generalization and the robustness of the proposed method.

---

### Official Review · Reviewer_yo1C · 2024-11-04

**Soundness:** 3
**Presentation:** 3
**Contribution:** 3
**Rating:** 6
**Confidence:** 2

**Summary:**

This paper introduces Gain Map-based Inverse Tone Mapping (GM-ITM), focusing on estimating the Gain Map (GM) for SDR images rather than directly converting to HDR. The proposed dual-branch network, GMNet, effectively combines local and global information for accurate GM prediction. Extensive experiments on both synthetic and real-world datasets demonstrate GMNet’s advantages over existing HDR methods, showing improved performance in both quantitative and qualitative metrics. The paper also contributes new datasets for GM-ITM research, supporting future advancements in HDR image processing.

**Strengths:**

1. **Innovative Task Definition**: The paper introduces a novel task, Gain Map-based Inverse Tone Mapping (GM-ITM), which focuses on GM estimation rather than direct HDR prediction, leveraging a unique double-layer HDR format for enhanced up-conversion.

2. **Effective Network Design**: The proposed GMNet utilizes a dual-branch structure with Local Contrast Restoration (LCR) and Global Luminance Estimation (GLE) branches, effectively capturing both pixel-level and image-level information for accurate GM prediction.

3. **Comprehensive Dataset Contribution**: The authors provide both synthetic and real-world SDR-GM datasets, which are diverse and well-suited for evaluating GM-ITM, fostering further research in this field.

4. **Strong Experimental Validation**: Extensive quantitative and qualitative experiments demonstrate GMNet’s superiority over existing HDR-related methods, showcasing its potential in real-world applications.

**Weaknesses:**

1. **Limited Error Analysis**: The paper provides little insight into errors GMNet might produce in challenging conditions (e.g., extreme lighting, high contrast), which is crucial for understanding its limitations.

2. **Model Size Trade-Offs**: An ablation study on model size versus performance could reveal if a smaller GMNet version maintains accuracy with reduced resources, benefiting applications needing speed-accuracy balance.

**Questions:**

1. **Insufficient Dataset Generation Information**: The process and diversity of the synthetic and real-world datasets are not fully detailed.
2.  **Lack of Real-Time Performance Evaluation**: The paper does not address GMNet’s computational efficiency or real-time processing capabilities, which are critical for practical HDR applications, especially on mobile devices.
3. **Scalability of the Model**: It’s unclear if GMNet can scale effectively with higher-resolution images, which is essential as HDR media often demands large image resolutions for detail preservation.

---

> ### Author Response · Authors · 2024-11-21
>
> We appreciate the reviewer's valuable comments and suggestions, and we hope our responses could address the concerns.
>
> **1. Insight into challenging conditions**
>
> Figure 10 in the Appendix demonstrates the superiority of our method in reconstructing high-contrast night scenes. To enable more in-depth analysis in challenging conditions, we conducted the following qualitative experiments.
>
> [Figure A] https://picx.zhimg.com/80/v2-f7a1fc6a62f5bf91169047c041d40f49.png
>
> [Figure B] https://picx.zhimg.com/80/v2-9a63b9bf1e886282ae886f41fccfad9e.png
>
> As shown in Figure A, our method achieves superior performance in the sun region at extreme brightnesses. The HDCFM also achieves low errors, but with grid effect due to operator properties. The Figure B demonstrates superior performance of our method in challenging condition with complex textures.
>
>
>
> **2. Model size trade-offs**
>
> Keeping the network structure unchanged, we adjust the number of hidden layers to control the model size. The results of the ablation experiments on the synthetic dataset are shown below, and our implementation in the paper is 64 hidden layers.
>
> | Hidden Layers | Params↓ | PSNR-L↑ | SSIM-L↑ | PSNR-PQ↑ | SSIM-PQ↑ | HDR-VDP3↑ |
> | :-----------: | :-----: | :-----: | :-----: | :------: | :------: | :-------: |
> |      32       |  0.46M  | 42.7029 | 0.9972  | 46.7620  |  0.9997  |  9.9319   |
> |      48       |  1.03M  | 43.3299 | 0.9973  | 47.4524  |  0.9998  |  9.9408   |
> |      64       |  1.83M  | 43.5510 | 0.9977  | 47.6256  |  0.9998  |  9.9477   |
>
> The experimental results show that reducing model parameters does not lead to significant performance degradation, especially in SSIM-L, SSIM-PQ and HDR-VDP3 that measure perception quality, verifying that our method can maintain accuracy with reduced resources.

---

> ### Author Response · Authors · 2024-11-21
>
> **3. More dataset information**
>
> In the Appendix of our paper, we present the diversity of the real-world dataset in Figure 7 and Figure 8. To address the reviewer's concerns, we demonstrate the diverse scenes and thumbnails of the synthetic dataset as follows.
>
> [Figure C] https://picx.zhimg.com/80/v2-c9660565e0e70b53d24970ceafcdb691.png
>
> [Figure D] https://picx.zhimg.com/80/v2-f0c2500d31685c44d74b83eca67fd82d.png
>
> The SDR-GM pairs of the real-world dataset are directly derived from the mobile device. The SDR images in the synthetic dataset are degraded from HDRTV frames. Specifically, we first roll off the input HDR by EETF and transfer it to the P3 gamut after linearization, then conduct extended Reinhard tone mapping, clip it under 100 nit, and finally store it in uint8 for quantization. After getting the SDR images, the detailed formation pipeline of the GM can be found in Section A in the Appendix. For more detailed information, we will release our codes and datasets upon acceptance of the paper.
>
>
>
>
> **4. Real-Time performance evaluation**
>
> To evaluate the computational efficiency and real-time processing capability of our method, we conduct comparison experiments and add three more baselines [1] [2] [3]. The runtime is evaluated in NVIDIA A100 as the average of 100 trials on the real-world dataset in the resolution of $4096\times3072$.
>
> |   ㅤMethodㅤ   | Params↓ |  ㅤRuntime↓ㅤ  |   ㅤFLOPs↓ㅤ   |   PSNR-L↑   |  PSNR-PQ↑   | $ΔE_{ITP}$↓ |
> | :---------: | :-----: | :--------: | :--------: | :---------: | :---------: | :---------: |
> |    KUNet    |  1.08M  |  621.70ms  |  7534.50G  |   33.3256   |   33.7895   |   9.0578    |
> |   HDRUNet   |  1.58M  |  613.89ms  |  4161.75G  |   33.1686   |   37.3966   |   5.7358    |
> |    HDCFM    |  0.10M  |  497.05ms  |  127.08G   |   32.7963   |   38.1894   |   6.2957    |
> |    FMNet    |  1.24M  |  304.13ms  |  4147.13G  |   32.6247   |   39.7632   |   4.5544    |
> | *DCDR-UNet* |  1.26M  | 1085.91ms  |  4502.77G  |   33.5421   |   38.3252   |   4.8645    |
> | *EPCE-HDR*  | 31.02M  | 23992.84ms | 326412.14G |   32.7616   |   31.1207   |   10.4322   |
> |  *ITM-LUT*  |  0.57M  |  19.81ms   |   59.16G   |   31.6677   |   37.9021   |   5.4299    |
> |    Ours     |  1.83M  |  75.26ms   |  1112.18G  | **33.9490** | **40.2088** | **4.0260**  |
>
> As can be seen, benefiting from the simple form of the target GM, our method achieves the second fastest runtime. While the look-up-table-based solution ITM-LUT is faster, its performance on image quality metrics is much lower.
>
> [1] Kim, Joonsoo, et al. "DCDR-UNet: Deformable Convolution Based Detail Restoration via U-shape Network for Single Image HDR Reconstruction." *Proceedings of the IEEE/CVF Conference on Computer Vision and Pattern Recognition*. 2024.
>
> [2] Tang, Jiaqi, et al. "High Dynamic Range Image Reconstruction via Deep Explicit Polynomial Curve Estimation." *ECAI 2023*. IOS Press, 2023. 2330-2337.
>
> [3] Guo, Cheng, et al. "Redistributing the Precision and Content in 3D-LUT-based Inverse Tone-mapping for HDR/WCG Display." *Proceedings of the 20th ACM SIGGRAPH European Conference on Visual Media Production*. 2023.
>
>
>
> **5. Scalability of the model**
>
> It needs to be clarified that the experiments in our paper are all performed on high-resolution images, where the resolution of the synthetic dataset is $3840\times2160$ and the resolution of the real-world dataset is $4096\times3072$. Therefore, the GMNet can scale effectively with high-resolution images in most cases.

---

> > ### Comment · Reviewer_yo1C · 2024-11-27
> >
> > Thank you for your response. You have already answered my question. I correspondingly increased my score.

---

> > > ### Author Response · Authors · 2024-11-28
> > >
> > > Thank you for raising the score.
> > >
> > > We truly appreciate the time and effort you have dedicated to reviewing our work and providing valuable feedback. Your comments and suggestions have been very insightful and helpful in improving the quality of our paper. Thank you again for your support and encouragement!

---

### Author Response · Authors · 2024-11-25
**General Response**

We sincerely thank the reviewers for their valuable comments and suggestions, and we hope our responses adequately address your concerns. The revised version of the manuscript has been uploaded. Additionally, we are happy to provide further details on any aspects of our responses that may require additional clarification or elaboration.

Once again, we appreciate the reviewers’ time and insightful feedback and look forward to receiving further input.

---

### Meta-Review · Area_Chair_wnbi · 2024-12-20

**Metareview:**

The paper presents a novel GM-ITM task for HDR imaging. It claims GMNet can capture local & global info for accurate GM estimation, with datasets supporting further research. Findings show GMNet's superiority. Strengths include the innovative task, the well-designed GMNet, valuable datasets, and extensive experiments. Weaknesses are a lack of error analysis in complex conditions, incomplete model size trade-offs, insufficient dataset details, and inadequate evaluation of real-time performance & scalability. Reasons for considering acceptance are novelty and good performance, but weaknesses need addressing for better scientific rigor and practicality.

**Additional Comments On Reviewer Discussion:**

The reviewers raised multiple important points. Reviewer yo1C asked for more error analysis in extreme conditions. The authors provided qualitative experiments (Figure A & B in Appendix), yet a more comprehensive quantitative analysis would better clarify limitations. For dataset generation, Reviewer nA6K sought details. The authors presented info on dataset diversity and the generation process, but deeper discussions on representativeness could be useful. Reviewer cLvc questioned computational complexity. The authors added runtime comparison experiments, though further exploration of practical usage impact is needed. Regarding novelty, Reviewer 6tzz had concerns. The authors argued and added discussions, but clearer establishment in HDR field is required. Overall, the comments are positive。

---

### Decision · Program_Chairs · 2025-01-22

Accept (Poster)